# Planning Goals for Exploration

**Edward S. Hu     Richard Chang     Oleh Rybkin     Dinesh Jayaraman**
GRASP Lab, Department of CIS, University of Pennsylvania
`{hued, huangkun, oleh, dineshj}@seas.upenn.edu`

## Abstract

Dropped into an unknown environment, what should an agent do to quickly learn about the environment and how to accomplish diverse tasks within it? We address this question within the goal-conditioned reinforcement learning paradigm, by identifying how the agent should set its goals at training time to maximize exploration. We propose "Planning Exploratory Goals" (PEG), a method that sets goals for each training episode to directly optimize an intrinsic exploration reward. PEG first chooses goal commands such that the agent's goal-conditioned policy, at its current level of training, will end up in states with high exploration potential. It then launches an exploration policy starting at those promising states. To enable this direct optimization, PEG learns world models and adapts sampling-based planning algorithms to "plan goal commands". In challenging simulated robotics environments including a multi-legged ant robot in a maze, and a robot arm on a cluttered tabletop, PEG exploration enables more efficient and effective training of goal-conditioned policies relative to baselines and ablations. Our ant successfully navigates a long maze, and the robot arm successfully builds a stack of three blocks upon command. Website:`https://sites.google.com/view/exploratory-goals`

## 1 Introduction

Complex real-world environments such as kitchens and offices afford a large number of configurations. These may be represented, for example, through the positions, orientations, and articulation states of various objects, or indeed of an agent within the environment. Such configurations could plausibly represent desirable goal states for various tasks. Given this context, we seek to develop intelligent autonomous agents that, having first spent some time exploring an environment, can afterwards configure it to match commanded goal states.

The goal-conditioned reinforcement learning paradigm (GCRL) (Andrychowicz et al., 2017) offers a natural framework to train such goal-conditioned agent policies on the exploration data. Within this framework, we seek to address the central problem: **how should a GCRL agent explore its environment during training time so that it can achieve diverse goals revealed to it only at test time?** This requires efficient unsupervised exploration of the environment.

Exploration in the GCRL setting can naturally be reduced to the problem of setting goals for the agent during training time; the current GCRL policy, commanded to the right goals, will generate exploratory data to improve itself (Ecoffet et al., 2021; Nair et al., 2018b). Our question now reduces to **the goal-directed exploration problem: how should we choose exploration-inducing goals at training time?**

Prior works start by observing that the final GCRL policy will be most capable of reaching familiar states, encountered many times during training. Thus, the most direct approach to exploration is to set goals in sparsely visited parts of the state space, to directly expand the set of these familiar states (Ecoffet et al., 2021; Pong et al., 2019; Pitis et al., 2020). While straightforward, this approach suffers from several issues in practice. First, the GCRL policy during training is not yet proficient at reaching arbitrary goals, and regularly fails to reach commanded goals, often in uninteresting ways that have low exploration value. For example, a novice agent commanded to an unseen portion of a maze environment might respond by instead reaching a previously explored part of the maze, encountering no novel states. To address this, prior works (Pitis et al., 2020; Bharadhwaj et al., 2021) set up additional mechanisms to filter out unreachable goals, typically requiring additional

hyperparameters. Second, recent works (Ecoffet et al., 2021; Yang et al., 2022; Pitis et al., 2020) have observed improved exploration in long-horizon tasks by extending training episodes. Specifically, rather than resetting immediately after deploying the GCRL policy to a goal, these methods launch a new exploration phase right afterwards, such as by selecting random actions (Pitis et al., 2020; Kamienny et al., 2022) or by maximizing an intrinsic motivation reward (Guo et al., 2020). In this context, even successfully reaching a rare state through the GCRL policy might be suboptimal; many such states might be poor launchpads for the exploration phase that follows. For example, the GCRL policy might end up in a novel dead end in the maze, from which all exploration is doomed to fail.

To avoid these shortcomings and focus exploration on the most promising parts of the environment, we propose to leverage planning with world models in a new goal-directed exploration algorithm, PEG (short for "Planning Exploratory Goals"). **Our key idea is to optimize directly for goal commands that would induce high exploration value trajectories**, cognizant of current shortcomings in the GCRL policy, and of the exploration phase during training. Note that this *does not* mean merely commanding the agent to novel or rarely observed states. Instead, PEG commands might be to a previously observed state, or indeed, even to a physically implausible state (see Figure 1). PEG only cares that the command will induce the chained GCRL and exploration phases together to generate interesting training trajectories, valuable for policy improvement.

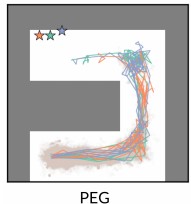 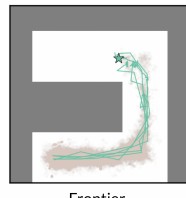

PEG             Frontier

Figure 1: PEG exploration in a U-maze. Brown background dots: explored states, $\star$: commanded goals, colored lines: resulting paths. (Left) PEG optimizes directly for exploration, even setting unseen goals, and achieving farther paths. (Right) Setting goals at the frontier of the seen state distribution yields less exploration.

Our key contributions are as follows. We propose a novel paradigm for goal-directed exploration by directly optimizing goal selection to generate trajectories with high exploration value. Next, we show how learned world models permit an effective implementation of goal command planning, by adapting planning algorithms that are often used for low-level action sequence planning. We validate our approach, PEG, on challenging simulated robotics settings including a multi-legged ant robot in a maze, and a robot arm on a cluttered tabletop. In each environment, PEG exploration enables more efficient and effective training of generalist GCRL policies relative to baselines and ablations. Our ant successfully navigates a long maze, and the robot arm successfully builds a stack of three blocks.

## 2   PROBLEM SETUP AND BACKGROUND

We wish to build agents that can efficiently explore an environment to autonomously acquire diverse environment-relevant capabilities. Specifically, in our problem setting, the agent is dropped into an unknown environment with no specification of the tasks that might be of interest afterwards. Over episodes of unsupervised exploration, it must learn about its environment, the various "tasks" that it affords to the agent, and also how to perform those tasks effectively. We focus on goal state-reaching tasks. After this exploration, a successful agent would be able to reach diverse previously unknown goal states in the environment upon command.

To achieve this, our method, **planning exploratory goals (PEG)**, focuses on improving unsupervised exploration within the goal-conditioned reinforcement learning framework. We set up notation and preliminaries below, before discussing our approach in section 3.

**Preliminaries.** We formalize the unsupervised exploration stage within a goal-conditioned Markov decision process, defined by the tuple $(\mathcal{S}, \mathcal{A}, \mathcal{T}, \mathcal{G})$. The unsupervised goal-conditioned MDP does not contain the test-time goal distribution, nor a reward function. At each time $t$, a goal-conditioned policy $\pi^G(\cdot \mid s_t, g)$ in the current state $s_t \in \mathcal{S}$, under goal command $g \in \mathcal{G}$, selects an action $a_t \in \mathcal{A}$, and transitions to the next state $s_{t+1}$ with probability $\mathcal{T}(s_{t+1}|s_t, a_t)$. To enable handling of the most broadly expressive goal-reaching tasks, we set the goal space to be identical to the state space: $\mathcal{G} = \mathcal{S}$, i.e., every state maps to a plausible goal-reaching command.

In our setting, an agent must learn a goal-conditioned reinforcement learning (GCRL) policy $\pi^G(\cdot \mid s, g)$ as well as collect exploratory data to train this policy on. Thus, as motivated in section 1, it

is critical to pick good goal commands during training because doing so generates useful data to improve the policy's ability to achieve diverse goals.

Critical to our method's design is the specific mechanism through which goal commands influence the exploration data. We follow the recently popular "Go-explore" procedure (Ecoffet et al., 2021; Pislar et al., 2022; Tuyls et al., 2022) for exploration-intensive long-term GCRL settings. Every training episode starts with the selection of a goal command $g$. Then, a goal-directed "**Go-phase**" starts, where the GCRL policy $\pi^G$ is executed, conditioned on $g$, producing a final state $s_T$ after $T$ steps. Immediately, an "**Explore-phase**" commences, with an undirected exploration policy $\pi^E(\cdot \mid s)$ taking over for the remaining $T_E$ timesteps. For example, $\pi^E(\cdot \mid s)$ could be an RL policy trained to maximize an intrinsic exploration reward $r^E$ (Bellemare et al., 2016; Pathak et al., 2017; Burda et al., 2019; Sekar et al., 2020). See Appendix D.1 for detailed pseudocode. This structure of training episodes has been shown to result in richer exploration (Pislar et al., 2022). Intuitively, the goal-conditioned policy efficiently brings an agent to an interesting goal, and the undirected exploration policy efficiently explores around the goal. However, Go-explore prescribes no general way to select goals that induce exploration.

In this paper, we propose PEG , a general goal command selection objective for Go-Explore-style unsupervised exploration. PEG uses a learned world model, making it natural to implement it alongside model-based RL approaches that already train such a model. We implement PEG exploration for "latent explorer achiever" (LEXA) (Mendonca et al., 2021), a goal-conditioned MBRL framework. We briefly summarize the LEXA framework below and refer the reader to Appendix D for details. LEXA learns the following components:

$$
\begin{aligned}
&\text{World model:} && \widehat{\mathcal{T}}_\theta(s_t|s_{t-1}, a_{t-1}) && && \\
&\text{Exploration policy:} && \pi_\theta^E(a_t|s_t) && \text{Goal Reaching policy:} && \pi_\theta^G(a_t|s_t, g) && (1) \\
&\text{Exploration value:} && V_\theta^E(s_t) && \text{Goal Reaching value:} && V_\theta^G(s_t, g) &&
\end{aligned}
$$

The world model is trained as a variational recurrent state space model following Hafner et al. (2020). The explorer and goal reaching policies are trained with the model-based on-policy actor-critic Dreamer algorithm (Hafner et al., 2021). Both policies are trained purely in imagination through world model "rollouts", i.e., the world model acts as a learned simulator in which to run entire training episodes for training these policies. The explorer reward

---

**Algorithm 1** LEXA Training Loop

1: **Input:** $\pi^G, \pi^E$, world model $\widehat{\mathcal{T}}$, rewards $r^G, r^E$
2: $\mathcal{D} \leftarrow \{\}$ Initialize buffer.
3: **for** Episode $i = 1$ to $N_{\text{train}}$ **do**
4: $\quad$ $\tau \leftarrow$ GoalDirectedExploration($\dots$)
5: $\quad \mathcal{D} \leftarrow D \cup \tau$
6: $\quad$ Update model $\widehat{\mathcal{T}}$ with $(s_t, a_t, s_{t+1}) \sim D$
7: $\quad$ Update $\pi^G$ in imagination with $\widehat{\mathcal{T}}$ to maximize $r^G$
8: $\quad$ Update $\pi^E$ in imagination with $\widehat{\mathcal{T}}$ to maximize $r^E$

---

is the Plan2Explore (Sekar et al., 2020) disagreement objective, which incentivizes reaching states that cause an ensemble of world models to disagree amongst themselves. The goal reaching reward $r^G$ is the self-supervised temporal distance objective (Mendonca et al., 2021), which rewards the policy for minimizing the number of actions between the current state and a goal state.

See Algorithm 1 for high level pseudocode of the training process. Highlighted in red is the goal-directed exploration strategy (Algorithm 2). LEXA explores by sampling random goals to reach or by running the exploration policy. As we will see in the next section, PEG serves as a drop-in replacement for LEXA's random goal sampling by sampling goals optimized for Go-explore exploration.

---

**Algorithm 2** LEXA Goal Sampling

1: **function** GOALDIRECTEDEXPLORATION($\dots$)
2: $\quad$ **if** episode $i \% 2 = 0$ **then**
3: $\quad\quad$ Sample random goal $g$ from buffer.
4: $\quad\quad$ Collect trajectory $\tau$ with $\pi^G(\cdot \mid \cdot, g)$
5: $\quad$ **else**
6: $\quad\quad$ Collect trajectory $\tau$ with $\pi^E$
7: **return** $\tau$

---

## 3 PLANNING GOALS FOR EXPLORATION

Having set up the preliminaries, we return to the core question of this paper: how should we pick exploration-inducing goals to help acquire diverse skills? We have argued above that the answer is not to pick goals at the frontier of previously explored states, as in prior work (Yamauchi, 1998; Pitis et al., 2020; Bharadhwaj et al., 2021). Instead, at the beginning of each training episode, we propose to optimize directly for goals that would lead to the highest exploration rewards during the episode's

Explore-phase:

$$\max_g \mathbb{E}_{p_{\pi^G(\cdot|\cdot,g)}(s_T)} \left[ V^E(s_T) \right], \tag{2}$$

$$\text{where} \quad V^E(s_T) = \mathbb{E}_{\pi^E} \left[ \sum_{t=T+1}^{T+T_E} \gamma^{t-T-1} r_t^E \right], \tag{3}$$

and where $p_{\pi^G(\cdot|\cdot,g)}(s_T)$ is the distribution of terminal states generated when $\pi^G$ runs for $T$ steps conditioned on goal $g$. $V^E$, which we call *exploration value*, is the expected discounted intrinsic motivation return of the exploration policy, when launched from state $s_T$. Please refer to Figure 2 for a detailed visualization.

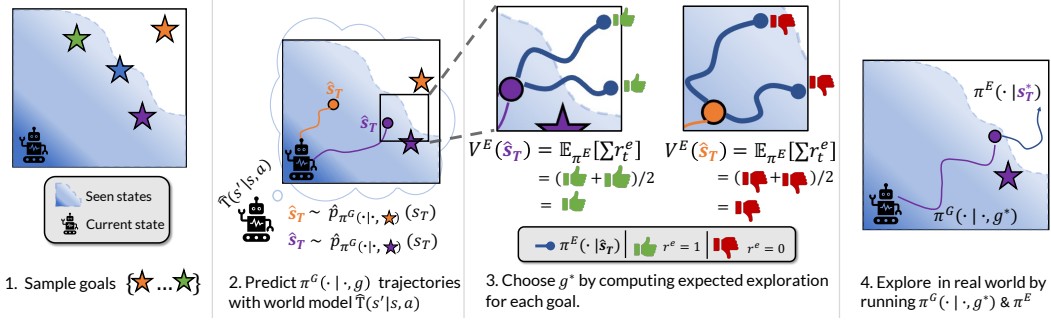

Figure 2: PEG proposes goals for Go-explore style exploration. 1) We start by sampling goal candidates $g_1 \ldots g_N$. 2) For each goal $g$, we execute the goal-conditioned policy $\pi^G$ in the world model $\widehat{\mathcal{T}}$ to generate trajectories. 3) We score each goal by seeing how useful their trajectories are for launching an exploration policy. 4) Explore using the best $g$.

Importantly, this objective involves computing the *expected* exploration value of terminal states $s_T$. The exploration value measures how interesting will the Explore-phase be as measured by intrinsic motivation rewards. Next, note that the expectation in Equation (2) is over the terminal states induced by the current GCRL policy. This way, it accounts for the capabilities of the current GCRL policy, regardless of whether it is a novice or an expert at achieving the selected goal $g$. If we had instead sought to optimize for goals $g$ with high exploratory value, in similar spirit to prior works (Pong et al., 2019; Pitis et al., 2020; Ecoffet et al., 2021), the objective of Equation (2) would instead be replaced by $V^E(g)$, the exploration value of the goal, which is naively agnostic to the capabilities of the GCRL policy. As we have argued above, this would then require additional mechanisms to account for reachability of the specified goal etc., bringing new hyperparameters and related overheads. Instead, PEG offers a unified and compact solution. In plain English, "first command goals that lead the goal policy to states that have high future exploration potential, then explore."

The objective in Equation (2) is not straightforward to compute, because it relies on the terminal state distribution induced by the goal-conditioned policy $p_{\pi^G(\cdot|\cdot,g)}(s_T)$. This distribution can change rapidly throughout training, as the policy evolves. To explore well, it is important to use an up-to-date estimate. We leverage the learned world model $\widehat{\mathcal{T}}$ to achieve this: $\hat{p}_{\pi^G(\cdot|\cdot,g)}(\tau)$ can be represented as the product of the transition and goal-conditioned policy distributions.

$$\hat{p}_{\pi^G(\cdot|\cdot,g)}(\tau) = p(s_0) \left[ \prod_{t=1}^{T} \widehat{\mathcal{T}}(s_t \mid s_{t-1}, a_{t-1}) \pi^G(a_{t-1} \mid s_{t-1}, g) \right] \tag{4}$$

Now, we may approximate the expectation over the marginal $p_{\pi^G(\cdot|\cdot,g)}(s_T)$ in the objective Equation (2) by sampling trajectories $\tau$ from the learned world model and using the last state $s_T$:

$$\mathbb{E}_{\hat{p}_{\pi^G(\cdot|\cdot,g)}(s_T)}[V^E(s_T)] \approx \frac{1}{K} \sum_k V_\theta^E(s_T^k) \qquad \text{where } s_T^k \sim \hat{p}_{\pi^G(\cdot|\cdot,g)}(\tau) \tag{5}$$

This permits on-the-fly generation of an up-to-date estimate of our objective from Equation (2), using a recent goal-conditioned RL policy. Then, we can optimize over goals $g$ using sample-based

optimization. Following Equation (5), to evaluate a goal $g$, we condition the goal-conditioned policy on $g$, and roll it out for $K$ trajectories $\tau_k$ within the learned world model. We then estimate the terminal state exploration value for each trajectory with the learned exploration value function $V_\theta^E(s_T^k)$ (Equation (1)), and average the estimates.

For optimizing the goal variable $g$, we use model predictive path integral control (Williams et al., 2015) (MPPI, see Appendix E for details). In Appendix E, we show that the specific choice of optimizer is not critical and other optimizers like cross entropy method (CEM) (De Boer et al., 2005) also work well. More broadly, while these sampling-based optimizers are popular for optimizing action trajectories with dynamics models (Nagabandi et al., 2019; Chua et al., 2018; Ebert et al., 2018; Hafner et al., 2020), PEG uses them instead for optimizing goal commands; this lower-dimensional search space enables easy optimization, independent of the task horizon, permitting handling long-term tasks.

Equipped with this goal setting procedure, we can now train a goal-conditioned policy by replacing the LEXA goal sampling highlighted in red in Algorithm 1 with our proposed PEG goals and Go-explore episodes. Algorithm 3 shows PEG pseudocode.

---

**Algorithm 3** PEG Goal Sampling

1: **function** GOALDIRECTEDEXPLORATION(. . .)
2:      $g \leftarrow$ Optimize Equation (5) with MPPI
3:      $\tau \leftarrow$ *GO-EXPLORE*$(g, \pi^G, \pi^E)$
4: **return** $\tau$

---

**Implementation Details.** We implement PEG on top of authors' code from DreamerV2 (Hafner et al., 2021), LEXA (Mendonca et al., 2021), and MPPI implementation from Rybkin et al. (2021). We used the default hyperparameters for training the world model, policies, value functions, and temporal reward functions. For PEG, we tried various values of $K$ for simulating trajectories of $\pi^G$ for each goal and found $K = 1$ to be sufficient. We use the same Go-explore mechanism across all goal-setting methods: the Go and Explore phases' time limits are set to half of the maximum episode length for all environments, while non-Go-explore baselines use the full episode length for exploration. See Appendix E for details on hyperparameters, runtime, and resource usage.

## 4 EXPERIMENTS

Our experiments evaluate goal-reaching tasks that require non-trivial exploration and skill acquisition to solve. We aim to answer: **(1)** Does PEG result in better exploration and downstream goal-reaching behavior? **(2)** What qualitatively distinguishes PEG goals and exploration from previous goal-directed exploration methods? **(3)** What components of PEG contribute to its success?

### 4.1 SETUP

We evaluate PEG and other goal-conditioned RL agents on four different continuous-control environments, described below. For each environment, we define an evaluation set of goals — as a general principle, we pick evaluation goals in each environment that require extensive exploration in order for the agent to learn a successful evaluation goal reaching policy. These goals are very relevant in long-horizon tasks, as long-horizon goals tend to require extensive exploration to gather the necessary data for training. Recall that we assume the fully unsupervised exploration setting; the agent does not have access to the evaluation set during training time. So, for all the agent knows at training time, any environment state could plausibly be a test goal. During evaluation, we run 10 goal-reaching episodes to account for environmental noise[1] for each goal, and report the mean success rate and standard error for each environment. All methods are run with 10 seeds. The success metric $d(s, g)$ used for evaluation is an indicator function checking if any state in the episode is within $\epsilon$ distance of the goal state. See Figure 3 for images of each environment, and Appendix C for more details.

- **Point Maze:** The agent, a 2D point, is noisily initialized in the bottom left. At test time, we evaluate it on reaching the right corner. The 2-D state and action space correspond to the planar position and velocity respectively. The episode length is 50 steps. The top right corner is the hardest to reach, so we set this to be the test-time goal.

- **Walker:** In this environment from Tassa et al. (2018), a 2D 2-legged robot moves on a flat plane. The state space is 9-D and contains the walker's XZ position and joint angles. The action space is 6-D and represents joint torques. The episode length is 150 steps. To evaluate the ability to learn

---

[1]Our environments have deterministic transitions with noise in the initial state distribution.

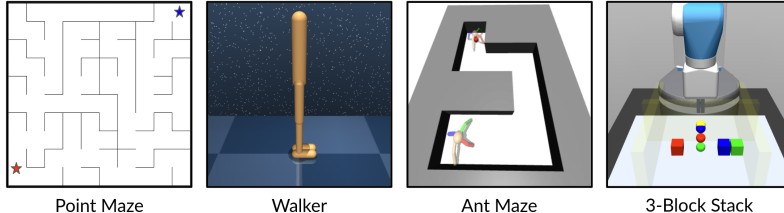

Point Maze      Walker      Ant Maze      3-Block Stack

Figure 3: We evaluate PEG across four diverse environments, selecting evaluation tasks to stress-test exploration. From left to right: Point Maze requires the agent to move from the bottom left to top right. Walker evaluates if the agent can walk to different positions on the plane. In Ant Maze, the ant is required to move to the farthest end of the maze. Finally, in 3-Block Stack, the robot must stack the blocks into the goal configuration (colored orbs).

different locomotion behaviors of varied difficulty, the test goal commands are standing states that are placed $\pm 7$, and $\pm 12$ meters from the initial pose.

• **Ant Maze:** We increased exploration difficulty in the Ant Maze environment from MEGA (Pitis et al., 2020) by placing one additional turn in the maze. Further, we set the GCRL goal space to the full 29-D ant state space, rather than reduce it to 2-D $xy$ position as in MEGA. The ant robot must learn complex 4-legged locomotion behavior and navigate around the hallways (see Figure 3). The episode length is 500 steps. We evaluate the ant on reaching 4 hard goals that are placed in the top left room, the furthest area from the start, and 4 intermediate goals in the middle hallway.

• **3-Block Stacking:** Here, a robot arm with a two-fingered gripper operates on a tabletop with 3 blocks and boundary walls. The state space is the gripper pose and object $xyz$ positions (14-D), and the action space is the gripper velocity and force (4-D). At evaluation time, the robot is commanded to stack three blocks into a tower configuration. To achieve this, the agent will need to have learned pushing, picking, and stacking, and discovered 3-block stacks as a possible configuration of the environment. The episode length is 150 timesteps. Note that 3+block stacking is a classic and open exploration challenge in robot RL; to our knowledge, all prior solutions assume expensive forms of supervision or computational resources to overcome the exploration challenge, such as demonstrations, curriculum learning, or billions of simulator samples (Ecoffet et al., 2021; Nair et al., 2018a; Lanier et al., 2019; Li et al., 2020), highlighting task difficulty.

## 4.2 BASELINES

PEG aims to improve goal-directed exploration by improving goal selection during GCRL. To specifically evaluate exploration, we implemented all baselines on top of LEXA (Mendonca et al., 2021), the backbone goal-conditioned MBRL framework by replacing the LEXA exploration logic (line 4 in Algorithm 1) with each method's proposed exploration strategy. See Appendix D for a detailed comparison between baselines.

Our first two baselines, Skewfit and MEGA, are alternative goal setting approaches: both follow Algorithm 1 and replace the LEXA exploration with their own goal sampling strategy for Go-Explore style exploration (see Algorithm 7 and Algorithm 8 for pseudocode). **Skewfit** (Pong et al., 2019) estimates state densities, and chooses goals from the replay buffer inversely proportional to their density, resulting in uniform state sampling. **MEGA** (Pitis et al., 2020) similarly uses kernel density estimates (KDE) of state densities, and chooses low density goals from the replay buffer. MEGA is close to the original Go-explore approach (Ecoffet et al., 2021); rather than Go-explore's hand-designed pseudocount metric, MEGA uses KDE to choose goals. MEGA relies on a heuristic Q-function threshold to prune unachievable low density goals since many rare states may not be achievable by $\pi^G$. This requires tuning and prior knowledge of the test goal to set.

We also run vanilla **LEXA** itself as a baseline. LEXA does not perform Go-explore style exploration, instead using an exploration policy for some exploration episodes and the goal-conditioned policy conditioned on random goals for others. In addition, we evaluate against a variant of LEXA that only collects data with the exploration policy. We denote this baseline Plan2Explore (**P2E**) after the intrinsic motivation policy it uses (Sekar et al., 2020). P2E trains an exploration policy to maximize

disagreement among an ensemble of world models. Notably, PEG and our other Go-Explore baselines all use the same P2E exploration algorithm during Explore-phase, which isolates the differences in exploration to the proposed goals of each method. See Appendix D for additional details including a side-by-side algorithm comparison, and where we verify that our MBRL versions of MEGA and Skewfit are sound and in fact better than the original model-free versions in Pitis et al. (2020).

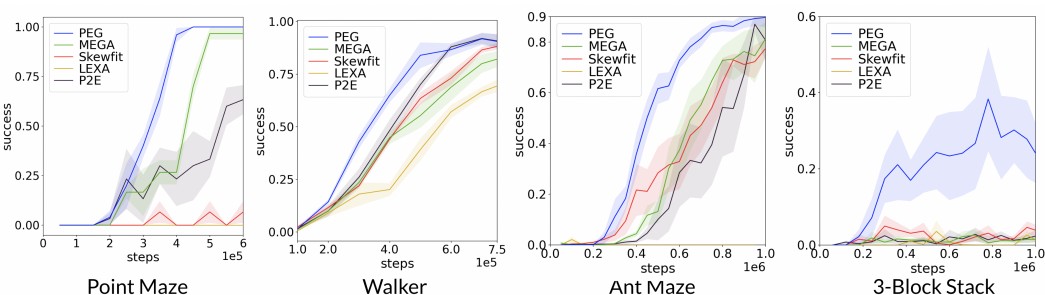

Figure 4: Performance of agents with different goal setting strategies. All methods are run with 10 seeds. PEG outperforms all baselines, and its performance gain increases with environment difficulty.

## 4.3 RESULTS

**Test task success rates.** Figure 4 shows the evaluation performance across training time. PEG compares favorably to baselines both in final success rate and learning speed. PEG reaches near-optimal behavior earlier than any other baseline. PEG outperforms goal-setting baselines (MEGA and Skewfit) across all tasks, which shows that PEG goal setting is superior to alternative goal-setting strategies. On the hardest task, block stacking, PEG is the only method to achieve any significant non-zero task performance: PEG achieves about 30% success rate on this challenging exploration task, all other baselines are close to 0%.

Recall that MEGA picks minimum density goals from the replay buffer. As we have seen in Figure 1, picking goals from the replay buffer may result in suboptimal goals for exploration as there could exist goals outside the data that result in more exploration. Furthermore, the MEGA objective does not reason about temporally extended exploration that begins after the goal state is reached. Skewfit, which samples goal states uniformly by their density, may often sample uninteresting goals, thus slowing down exploration. In our upcoming qualitative analysis of goal selection behavior, we see these conceptual differences manifest in suboptimal goal picking in comparison to PEG .

Next, P2E has no Go-phase, and thus no goal-directed exploration, which limits its exploration horizon. Even so, it can sometimes perform better than MEGA and Skewfit (see Walker), both methods that employ the same exploration algorithm as P2E, just after a Go-phase. In other words, Go-explore by itself doesn't always improve exploration, and suboptimal goal-setting during "Go-phase" can in fact deteriorate exploration. LEXA performs worse than simple P2E, indicating that sampling random goals from the replay buffer does not improve exploration or training.

**Goal selection behaviors.** Next, Figure 5 visualizes the goal-picking and exploration behaviors for PEG and baselines. A trend across tasks is that PEG consistently picks goals (red points) beyond the frontier of seen data. In **Point Maze**, the middle right region forms a natural barrier for the exploration policy to overcome. The policy must find a rather tortuous path from the bottom left through the middle right hallways in order to reach the top right corner. As a result, top right states beyond the middle right region are very sparsely represented. With similar states in the replay buffer (blue dots), we find that PEG generates many more goals in the top right region than MEGA, thus accelerating exploration. Similarly, in **Walker**, PEG selects goals to the far left and right of the walker, where data support is the sparsest. MEGA fails to pick goals to the left.

In **Ant Maze** (Figure 5), we see that PEG explores the most deeply into the maze of all methods, at this halfway point. Counting the total number of state visitations in the farthest top-left reaches of the maze for each method, we find that PEG has ∼68K states whereas MEGA, the closest baseline, only has ∼3K. Looking at Figure 5, MEGA does pick a few "rare" goals in the top left, but $\pi^G$ has not visited the top left area, either because $\pi^G$ is not capable of reaching the rare goals and/or the

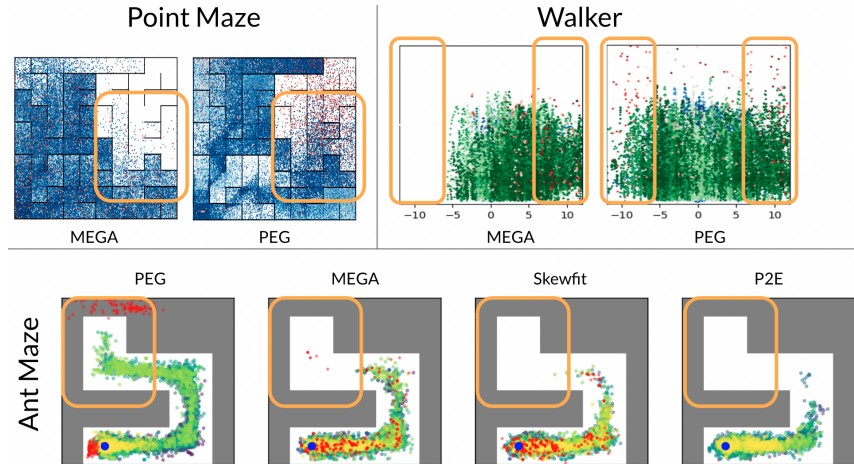

Figure 5: We plot the achieved states (blue / green dots) and goals (red dots) for PEG and baselines in the Point Maze (top left), Walker (top right), and Ant Maze (bottom) halfway through training. PEG picks goals in areas (orange boxes) that lead to more exploration whereas baselines do not. Walker and Ant states are high dimensional so we just plot their central $xz$ or $xy$ positions.

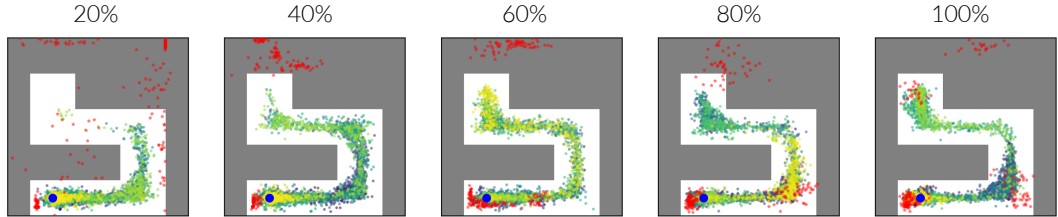

Figure 6: The evolution of achieved states (colored dots) and goals (red dots) of PEG throughout training in the Ant Maze. Lighter states are more recent. PEG samples goals to maximize exploration, sometimes picking goals that are outside the seen data or even physically implausible.

frontier. Skewfit samples no goals in the top left region, because it does not prioritize rare states. We further examine the training evolution of PEG goals in Ant Maze to analyze how they contribute to exploration. We plot the achieved states and goals proposed by PEG periodically throughout its training in Figure 6. PEG goals, optimized to induce the goal-conditioned policy to visit high-exploration value states, are not always predictable. PEG commonly chooses goals outside of the data distribution. For example, PEG chooses goals near the top wall for the entire training duration. We can also see it choosing physically impossible goals in the middle wall at 20%, but these goals disappear as the goal-conditioned policy and world model are updated over time.

In the **3-Block Stack** environment, as discussed above, PEG is the only method to train any meaningful policy for 3-block stacking. The space of configurations of the three blocks and the robot in this setting is extremely large. While all methods are able to explore and discover simpler skills such as picking and 2-block stacking, only PEG's exploration is comprehensive enough to also explore more "useful" and hard-to-reach behaviors such as 3-block stacking. See Appendix F.4 for success curves on intermediate goals like picking and 2-block stacking, and the website[2] for videos.

**Ablation Analysis.** What components of PEG are important to its performance? We run ablation studies in Ant Maze. First, we ablate PEG's goal sampling: Rand-Goal PEG replaces PEG's goal sampling with randomly sampled goals. In Seen-Goal PEG, we forego optimizing the goal with MPPI and simply choose goals from the replay buffer that score highly according to Equation (5). Go-Value PEG chooses to optimize the exploration rewards achieved during the imagined *Go-phase* instead of during the Explore-phase, by replacing $V^E(s_T)$ in Equation (5) with $\sum_{t=1}^{T} r_t^E$ — note that this still runs Explore after Go in the environment.

[2] https://sites.google.com/view/exploratory-goals

Next, we ablate PEG's Go-explore mechanism itself, by replacing the explorer policy with a random action policy (Rand-Explore PEG), and removing Explore-phase completely (No-Explore PEG).

Figure 7 shows the results. PEG is clearly better than all ablations, suggesting that all components of the method contribute to performance. Explore-phase is particularly important; No-Explore PEG achieves 0% success. Rand-Explore PEG performs poorly, showing the need for a learned Explore-phase policy like P2E. Next, Rand-Goal PEG, Seen-Goal PEG, and Go-Value PEG all perform worse than PEG; the specifics of the objective function for goal sampling defined in Equation (5), and its optimization procedure, are important.

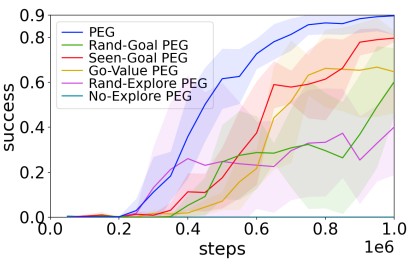

Figure 7: PEG ablations.

## 5   OTHER RELATED WORK

PEG is most closely related to the subfield of unsupervised exploration in RL, where no tasks are pre-specified during training. In this setting, a common approach is to define an intrinsic motivation reward correlated with exploration for RL (Schmidhuber, 2010; Pathak et al., 2017). Common forms of intrinsic reward incentivize visiting rare states with count-based methods (Poupart et al., 2006; Bellemare et al., 2016; Burda et al., 2019) or finding states that maximize prediction error (Oudeyer et al., 2007; Pathak et al., 2017; Henaff, 2019; Shyam et al., 2019; Sekar et al., 2020). Within PEG, we leverage an intrinsic motivation policy for the Explore-phase.

More specifically related to PEG, goal-directed exploration is a special class of unsupervised exploration, applicable in the challenging goal-conditioned RL (GCRL) setting. To develop generalist GCRL policies, these methods command the goal-conditioned policy towards exploratory goals at training time. Broadly, the idea is to pick goals that are difficult for the current policy, but not completely out of reach, i.e., within the "zone of proximal development." Within that broad framework, however, prior works propose many different metrics for goal choosing, such as learning progress (Baranes & Oudeyer, 2013; Veeriah et al., 2018), goal difficulty (Florensa et al., 2018), "sibling rivalry" (Trott et al., 2019), or value function disagreement (Zhang et al., 2020). Within this family, we have already discussed and compared against the most closely relevant methods MEGA and Skew-Fit (Pong et al., 2019; Pitis et al., 2020) (see Section 4.2 and Section 4.3) which command the agent to rarely seen states to increase the number of seen and familiar states, reasoning that this will enable broader expertise in the trained GCRL policies.

Closely related to unsupervised goal-directed exploration is unsupervised skill discovery, which aims to train a skill-conditioned policy Eysenbach et al. (2019); Sharma et al. (2019). Skill discovery objectives do not necessarily encourage exploration, and can even fall into local minima where the agent is penalized for exploration (Strouse et al., 2021; Campos et al., 2020). As such, these are usually weak baselines for exploration; for example, our goal-directed exploration baselines, LEXA and MEGA, both report clearly favorable results against such methods (Eysenbach et al., 2019; Hartikainen et al., 2020). See Appendix A for an extended discussion of related work.

## 6   CONCLUSIONS

We have presented PEG, an approach that sets goals to achieve unsupervised exploration in goal-conditioned reinforcement learning agents. PEG plans these exploratory goals through a world model, directly optimizing for the estimated exploration value of the training trajectory. While PEG performs better in our experiments than prior approaches across various environments, exploration for long-horizon tasks remains challenging, and PEG's explicit dependence on world model rollouts may be a particular weakness in such settings. Likewise, complex environments may permit extremely large goal spaces, where purely unsupervised exploration approaches like PEG are unlikely to succeed within reasonable time, and some information about the target task space may be required. We believe that the PEG principle of optimizing for goals predicted to induce highly exploratory policy behaviors will prove useful towards future efforts to overcome these longstanding exploration challenges.

## 7 ACKNOWLEDGEMENTS

This work was supported in part by ONR grant number N00014-22-1-2677 and gift funding from NEC Laboratories. The authors would like to thank the anonymous reviewers for their constructive feedback, as well as Yecheng Jason Ma and the other members of the Perception, Action, and Learning (PAL) research group at UPenn for useful discussions and general support.

## 8 REPRODUCIBILITY STATEMENT

To ensure reproducibility, we will release the codebase that contains our method, baselines, and environments. The supplementary contains details about hyperparameters, baselines, and architecture (see Appendix D and Appendix E). In addition, we have also provided pseudocode in Algorithm 3.

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

# Appendix

## Table of Contents

## A  EXTENDED RELATED WORK

Exploration is a very broad field, and we encourage readers to read surveys such as Yang et al. (2021). Below, we summarize various subfields of exploration and reinforcement learning relevant to PEG.

**Task-directed exploration.**    PEG reasons about the intrinsic motivation / exploration rewards to be gained by commanding a goal-conditioned task policy to various goals in an unsupervised RL setting. Related at a high level, supervised exploration approaches reason about task reward gains during exploration. (Chen et al., 2019; Osband et al., 2019) leverage uncertainty estimates about the reward to prioritize states for task-directed exploration. (Stratonovich & Grishanin, 1966) proposes "value of information" (VoI) to measure exploration from an information-theoretic perspective. The VoI describes the improvement in supervised task rewards of the current optimal action policy operating under a specified limit on state information. An RL agent may modulate exploration-vs-exploitation in supervised RL settings, by tuning this information limit: a higher limit leads to lower exploration and greater exploitation (Sledge & Príncipe, 2017; Cogliati Dezza et al., 2017).

**Unsupervised Exploration.** In unsupervised exploration, no information about the task (e.g. reward function or demonstrations) is known during training time. In this setting, a common approach is to define an intrinsic motivation reward correlated with exploration for RL (Schmidhuber, 2010; Pathak et al., 2017). Common forms of intrinsic reward incentivize visiting rare states with count-based methods Poupart et al. (2006); Bellemare et al. (2016); Burda et al. (2019) or finding states that maximize prediction error (Oudeyer et al., 2007; Pathak et al., 2017; Henaff, 2019; Shyam et al., 2019; Sekar et al., 2020). Within PEG, we leverage an intrinsic motivation policy for the Explore-phase.

**Goal-conditioned RL.** Goal-conditioned RL (GCRL) extends RL to the multitask setting where the policy is conditioned on a goal and is expected to achieve it. The goal space is commonly chosen to be the state space, although other options like desired returns or language inputs are possible. GCRL is challenging since the policy is required to learn the correct behavior for achieving each goal. This introduces two challenges - optimizing a goal-conditioned policy, and getting adequate data to support the optimization process.

Many GCRL methods focus on improving goal-conditioned policy optimization. Hindsight Experience Replay Andrychowicz et al. (2017) boosts training efficiency by relabeling failure trajectories as successful trajectories, and Chane-Sane et al. (2021) uses imagined subgoals to guide the policy search process. However, such methods are only useful if the data distribution is diverse enough to cover the space of desired behaviors and goals, and will still suffer in hard exploration environments. Therefore, GCRL is still predominantly constrained by exploration.

**Goal-directed Exploration.** Goal-directed exploration, which sets exploratory goals for the goal-conditioned policy to achieve, is often the exploration mechanism of choice for GCRL as it naturally reuses the goal-conditioned policy. It is important to note that GDE is not inherently tied to GCRL, and can be used in conventional RL to solve hard exploration tasks (Ecoffet et al., 2021; Guo et al., 2020; Hafner et al., 2022). In such cases, goals are usually picked through some combination of reward and novelty terms.

Prior works propose many different metrics for goal choosing, such as frontier-based (Yamauchi, 1998), learning progress (Baranes & Oudeyer, 2013; Veeriah et al., 2018), goal difficulty (Florensa et al., 2018), "sibling rivalry" (Trott et al., 2019), or value function disagreement (Zhang et al., 2020). Within this family, we have already discussed and compared against the most closely relevant methods MEGA and Skew-Fit (Pong et al., 2019; Pitis et al., 2020) (see Section 4.2 and Section 4.3) which command the agent to rarely seen states to increase the number of seen and familiar states, reasoning that this will enable broader expertise in the trained GCRL policies.

**Go-Explore.** The Go-Explore framework from (Ecoffet et al., 2021) performs goal-directed exploration by first rolling out the goal-conditioned policy (Go-phase) and then rolling out the exploration policy from the terminal state of the goal-conditioned phase (Explore-phase). Go-Explore does not prescribe a general goal setting method, instead opting for a hand-engineered novelty bonus for each task. GDE methods like MEGA (Pitis et al., 2020) use the Go-Explore mechanism to further boost exploration efficiency, but do not consider Go-Explore when designing their goal-setting objective. This results in suboptimal performance - MEGA goals chooses low density goals, but this may not necessarily mean they are good states to start the exploration phase from (e.g. dead-ends in a maze). PEG 's objective already takes the Go-Explore mechanism into account, choosing goals that maximize the exploration of the Explore phase and resulting in superior exploration.

## B EXTENDED LIMITATIONS AND FUTURE WORK

**Model-based RL vs. Model-free RL.** PEG is a model-based RL agent, which is more sample-efficient and computationally expensive than model-free counterparts. MBRL agents require more computation time and resources since they learn a world model. PEG trains a world model to train policies and value functions with imagined trajectories, as well as generate goals for exploration. A model-free version of PEG should be computationally and conceptually simpler, and we leave this for future work.

**Better world models.** PEG depends on a learned world model to plan goals, and may suffer in prediction performance and computation time for longer prediction horizons. An interesting direction

would be to combine PEG with temporally abstracted world models (Neitz et al., 2018; Jayaraman et al., 2019; Pertsch et al., 2020) to handle such cases. Furthermore, PEG should benefit with improved world model architectures such as Transformers (Micheli et al., 2023), and thus improving world modelling offers a straightforward path for extending PEG .

**Exploration beyond state novelty.** Further, while empirically PEG produces performant goal-conditioned policies, it only explores by maximizing novelty. In harder tasks it might be necessary to explicitly consider also exploring with "practice goals" that best improve the goal-conditioned policy.

Next, Pislar et al. (2022) suggests that generalizing the Go-Explore paradigm itself is beneficial. In Go-Explore, we assume the agent first performs a "Go-phase" for the first half of the episode, and then the "Explore-phase" for the remaining half. However, we can allow the agent to switch between the "Go-phase" and "Explore-phase" multiple times in an episode, and decide the phase lengths.

**Scaling PEG to harder tasks.** While we focus on purely unsupervised exploration, in practice some knowledge about the task is always available. For example, a human may give some rewards, a few demonstrations, or some background knowledge about the task. Leveraging such priors to focus exploration on the space of tasks preferred by the user would make GCRL and PEG more practical, especially in real world applications like robotics.

PEG currently is only evaluated in state space, and a potential extension would be to handle image observations by optimizing goals in a compact latent space. Such an extension should only require minor edits in the PEG code because PEG is extended from LEXA, an image-based RL agent that already learns such a latent space.

## C  ENVIRONMENTS

### C.1  POINT MAZE

The agent, a 2D point spawned on the bottom left of the maze, is evaluated on reaching the top right corner within 50 timesteps. The two-dimensional state and action space correspond to the planar position and velocity respectively. Success is based upon the L2 distance between the position of the agent and the goal being less than 0.15. The entire maze is approximately 10 x 10. This environment is taken directly from Pitis et al. (2020) with no modifications.

### C.2  WALKER

Next, the walker environment tests the locomotion behavior of a 2D walker robot on a flat plane. The agent is evaluated on its ability to achieve four different standing poses that are placed $\pm 7$, and $\pm 12$ meters in the $x$ axis away from the initial pose. For success, we check if the agent's $x$ position is within a small threshold from the goal pose's $x$ position. The state and goal space is 9 dimensional, containing the walker's $xz$ positions and joint angles. The environment code was taken from Mendonca et al. (2021).

### C.3  ANT MAZE

A high-dimensional ant robot needs to move through hallways from the bottom left to top left (see Fig. 3). This environment is a long-horizon challenge due to the long episode length (500 timesteps) and traversal distance. We evaluate the ant on reaching 4 hard goals at the top left room, and 4 intermediate goals in the middle hallway. Ant Maze has the highest dimensional state and goal spaces: 29 dimensions that correspond to the ant's position, joint angles, and joint velocities. The first three dimensions of the state and goal space represent the $xyz$ position. The next 12 dimensions represent the joint angles for each of the ant's four limbs. The remaining 14 dimensions represent the velocities of the ant in the x-y direction and the velocities of each of the joints. Ant Maze also has an 8 dimensional action space. The 8 dimensions correspond to the hip and ankle actuators of each limb of the ant. Success is based upon the L2 distance between the $xy$ position of the ant and the goal being less than 1.0 meters, which is approximately the width of a cell. The dimensions of the entire maze is approximately 6 x 8 meters. This environment is a modification from the Ant Maze environment in Pitis et al. (2020). We modified the environment so that the state and goal space now

includes the ant's $xyz$ positions along with the joint positions and velocities instead of just the $xy$ positions. We also added an additional room in the top left in order to evaluate against a harder goal.

## C.4 3-BLOCK STACK

A robot has to stack three blocks into a tower configuration. The state and goal space for this environment is 14 dimensional. The first five dimensions represent the gripper state and the other nine dimensions represent the $xyz$ positions of each block. The action space is 4 dimensional where the first three dimensions represent the $xyz$ movement of the gripper and the last dimension represents the movement of the gripper finger. Success is based upon the L2 distance between the $xyz$ position of all of object and the goal position of the corresponding object being less than 3cm. This environment is a modification from the FetchStack3 environment in Pitis et al. (2020).

## D BASELINES

Here, we first present pseudocode for all methods. We start by defining a generic MBRL training loop in Algorithm 4 that alternates between data collection (instantiated by specific methods) and policy and world model updates. Next, we define the pseudocode for PEG and all baselines below. To run a method, we simply drop in the data collection logic specified by the method

---

**Algorithm 4** MBRL Training

1: **Input:** goal / expl. policies $\pi^G$, $\pi^E$, world model $\widehat{\mathcal{T}}$, rewards $r^G$, $r^E$
2: $\mathcal{D} \leftarrow \{\}$ Initialize buffer.
3: **for** Episode $i = 1$ to $N_{\text{train}}$ **do**
4:     $\tau \leftarrow$ GoalDirectedExploration($\ldots$)
5:     $\mathcal{D} \leftarrow D \cup \tau$
6:     Update model $\widehat{\mathcal{T}}$ with $(s_t, a_t, s_{t+1}) \sim D$
7:     Update $\pi^G$ in imagination with $\widehat{\mathcal{T}}$ to maximize $r^G$
8:     Update $\pi^E$ in imagination with $\widehat{\mathcal{T}}$ to maximize $r^E$

---

into the MBRL training loop. For exploration, LEXA and P2E do not use Go-explore, while PEG, MEGA, and Skewfit do.

### D.1 GO-EXPLORE

Go-Explore (Ecoffet et al., 2021) proposes to switch between a goal-directed policy and exploration policy in a single episode to tackle exploration-intensive environments. Go-explore first stores states of "interest" as future goal states. Then, in a new episode, Go-explore chooses a goal $g$, and directs a goal-directed policy $\pi^G$ to achieve the goal (or upon timeout $T_{\text{Go}}$). Then, for the rest of the episode, an exploration policy $\pi^E$ is used. Intuitively, this results in frontier-reaching exploration (Yamauchi, 1998)

---

**Algorithm 5** Go-explore

1: **function** GO-EXPLORE($g, \pi^G, \pi^E$)
2:     $s_0 \leftarrow$ env.reset()
3:     $\tau \leftarrow \{s_0\}$
4:     **for** Step $t = 1 \ldots T_{\text{Go}}$ **do**
5:         $s_t \leftarrow$ env.step($\pi^G(s_{t-1}, g)$)
6:         $\tau \leftarrow \tau \cup \{s_t\}$
7:     **for** Step $t = 1 \ldots T_{\text{Explore}}$ **do**
8:         $s_t \leftarrow$ env.step($\pi^E(s_{t-1}, g)$)
9:         $\tau \leftarrow \tau \cup \{s_t\}$
10: **return** $\tau$

---

where the goal-directed policy efficiently traverses to the "frontier" of known states before doing exploratory behavior. Of particular importance to Go-Explore is how goal states are chosen - Ecoffet et al. (2021) does not prescribe a general approach for choosing goals. Instead, they choose goals by using task-specific psuedocount tables.

### D.2 LEXA

LEXA (Mendonca et al., 2021) trains a goal-conditioned policy with model-based RL. It trains two policies, an exploration policy and a goal-conditioned "achiever" policy. The original LEXA codebase was designed for image-based environments, so we implemented a state-based LEXA agent by modifying DreamerV2 (Hafner et al., 2021). To do so, we added the $\pi^G$ and $\pi^E$

---

**Algorithm 6** LEXA Goal Sampling

1: **function** GOALDIRECTEDEXPLORATION($\ldots$)
2:     **if** $i\%2 = 0$ **then**
3:         Sample random goal $g$ from buffer.
4:         Collect trajectory $\tau$ with $\pi^G(\cdot \mid \cdot, g)$
5:     **else**
6:         Collect trajectory $\tau$ with $\pi^E$
7: **return** $\tau$

---

policies, goal-conditioning logic, and the temporal distance reward network.

### D.3 ORIGINAL MEGA CODEBASE

We used the original MEGA codebase to train a MEGA and a Skewfit agent (Pitis et al., 2020; Pong et al., 2019) for 1 million environmental steps to solve the Pointmaze, Ant Maze, and 3-Block Stack tasks mentioned in Section 6. We used the original MEGA implementation with the same hyperparameters except for their goal-conditioned policy training scheme.

MEGA proposes Rfaab, a custom goal relabeling strategy to train the goal-conditioned policy. By default Rfaab relabels transitions with test time goals to train the goal-conditioned policy, which is not possible in our unsupervised exploration setup. Therefore, we changed the Rfaab hyperparameters to rfaab_0_4_0_5_1, which no longer grants access to test goals.

We find that both MEGA and Skewfit agent fail to scale to the harder environments (Ant Maze and 3-Block Stack) as seen in Figure 8. One reason could be that the MEGA codebase uses model-free RL, which is more sample inefficient. Therefore, we reimplemented MEGA and Skewfit with LEXA, the backbone MBRL agent that PEG uses as well.

|  | MEGA | Skewfit |
|---|---|---|
| Point Maze | 90% | 0% |
| Ant Maze | 0% | 0% |
| 3-Block Stack | 0% | 0% |

Figure 8: Success Rate of MEGA and Skewfit Agent

### D.4 MODEL-BASED MEGA

For model-based MEGA, we simply port over MEGA's KDE model. Conveniently, LEXA already trains a goal-conditioned value function, which we use for filtering out goals by reachability. We use the same KDE hyperparameters as the original MEGA paper. As seen in in Figure 4, our MEGA implementation gets non-trivial success rates, improving over the original MEGA baseline.

---
**Algorithm 7** MEGA Goal Sampling
---
1: **function** GOALDIRECTEDEXPLORATION(...)
2:     $g \leftarrow \min_{g \in \mathcal{D}} \widehat{p}(g)$
3:     $\tau \leftarrow$ *GO-EXPLORE*$(g, \pi^G, \pi^E)$
4: **return** $\tau$
---

### D.5 MODEL-BASED SKEWFIT

For Skewfit, we port over the Skewfit baseline in MEGA's codebase. The implementation is straightforward and similar to model-based MEGA. Instead of picking the minimum density goal, Skewfit picks goals inversely proportional to their density. As seen in in Figure 4, our Skewfit implementation gets non-trivial success rates.

---
**Algorithm 8** Skewfit Goal Sampling
---
1: **function** GOALDIRECTEDEXPLORATION(...)
2:     $g \sim \left\{ g \cdot \widehat{p}(g)^{-1} \mid g \in \mathcal{D} \right\}$
3:     $\tau \leftarrow$ *GO-EXPLORE*$(g, \pi^G, \pi^E)$
4: **return** $\tau$
---

### D.6 PLAN2EXPLORE

Our Plan2Explore baseline uses a Plan2Explore exploration policy to gather data for the world model and trains a goal-conditioned policy in imagination. Our LEXA implementation already uses Plan2Explore, so this variant simply requires us to only rollout the exploration policy in the real world.

---
**Algorithm 9** Plan2Explore
---
1: **function** GOALDIRECTEDEXPLORATION(...)
2:     Collect trajectory $\tau$ with $\pi^E$
3: **return** $\tau$
---

# E    IMPLEMENTATION DETAILS

## E.1    MPPI

MPPI is a popular optimizer for planning actions with dynamics models (Williams et al., 2015; Nagabandi et al., 2019). Instead of planning sequences of actions, which grows linearly with horizon, PEG only needs to "plan" with a goal variable, making the search space independent of the horizon. First, we present a high-level summary of MPPI, and refer interested readers to Nagabandi et al. (2019) for a more detailed look at MPPI.

The process begins by randomly sampling $N$ goal candidates $g$ from an initial distribution (see below for more details). These candidates are then evaluated as described in Figure 2 - we generate goal-conditioned trajectories for each goal $g_k$ through the world model, and compute the expected exploration value $V_k$ for each goal. This exploration value acts as the "score" for the goal candidate. Once we have scores for each goal candidate, a Gaussian distribution is fit according to the rule:

$$\mu_t = \frac{\sum_{k=0}^{N}(e^{\gamma \cdot V_k})(g_k)}{\sum_{k=0}^{N}(e^{\gamma \cdot V_k})} \tag{6}$$

where $\gamma$ is the reward weight hyperparameter. We then sample candidates from the computed Gaussian, and repeat the process for multiple iterations.

**MPPI initial distribution**    Throughout our paper, we initialize the MPPI optimizer with goal vectors sampled uniformly at random from the full goal space. One alternative would have been to restrict the initial candidates to lie in the agent's replay buffer, but we found in practice that this restriction made no difference and it was more convenient for implementation reasons to simply sample uniformly at random from the unrestricted goal space (same as the state space), e.g. using the joint limits of the various robot joints.

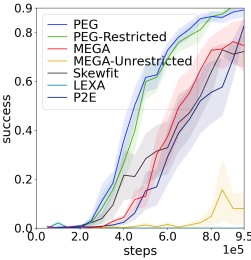

Figure 9: Performance of state space sampling variants on the Ant Maze. Removing state space sampling (PEG-Restricted) does not degrade PEG performance, and adding state space sampling (MEGA-Unrestricted) does not boost performance.

In Figure 9, we show results of running PEG with MPPI initialization from the replay buffer ("PEG-Restricted"), in the Ant Maze environment. Its performance is virtually indistinguishable from the original PEG method. This is to be expected, and is a sign that the optimizer is working well.

Could other methods benefit from sampling outside the replay buffer? In Figure 9 we allowed our strongest baseline MEGA to also sample goals from outside the replay buffer ("MEGA-Unrestricted"). However, these methods rely on heuristic objectives for selecting goals (like "low density" of the visited state distribution) which only approximate the correct goal-setting objective for states close to the replay buffer. Therefore, "MEGA-Unrestricted" is unable to sample effective goals outside the replay buffer, and fares very poorly. In contrast, PEG directly optimizes the exploration value of goal states, and permits choosing meaningful goals even in this unrestricted setting. This is an advantage, since it permits optimizing over a larger space of goal candidates, which leads to better goals.

## E.2 RUNTIME

| | Total Runtime (Hours) | Episodes | Episode Length | Seconds per Episode |
|---|---|---|---|---|
| Point Maze | 16 | 12000 | 50 | 4.8 |
| Walker | 16 | 5000 | 150 | 11.52 |
| AntMaze | 48 | 2000 | 500 | 86.4 |
| 3-Block Stack | 48 | 6666 | 150 | 25.9 |

Figure 10: Runtimes per experiment.

The runtime and resource usage between methods did not differ significantly, as everything is implemented on top of the backbone LEXA model-based RL agent. Note that model-free agents tend to run significantly faster than model-based agents, but fail to explore and optimize good policies. See Appendix D where we found model-free variants of MEGA and Skewfit failed to learn at all. Runtime is dominated by neural network updates of the policies and world model, not the goal selection routine defined by each method. Each seed was run on 1 GPU (Nvidia 2080ti or Nvidia 3090) and 4 CPUs, and required 11GB of GPU memory. See Figure 10 for training time.

| | Seconds / Episode |
|---|---|
| PEG: | 0.51 |
| MEGA: | 0.48 |
| SkewFit: | 0.46 |

Figure 11: Runtime for goal selection.

Next, we benchmarked the goal selection procedure for PEG and the other goal selection baselines in the block stacking environment and recorded the average wall clock time in Figure 11.

We can see that there is little difference in speed between methods, and the overhead introduced by PEG goal selection is minimal to LEXA, and competitive with other goal selection baselines. Because LEXA does not select goals, it finishes up to an hour earlier than goal setting methods in our experiments. As a sidenote, the times reported above are amortized over episodes, since we compute a set of 50 goals every 50 episodes for all methods, rather than computing 1 goal per episode.

## E.3 HYPERPARAMETERS

For parameter tuning, we used the default hyperparameters of the LEXA backbone MBRL agent (e.g. learning rate, MLP size, etc.) and kept them constant across all baselines. PEG only has 1 hyperparameter that requires tuning - the reward weight, which affects the MPPI update distribution. The higher the weight, the more we update the goal sampling distribution towards high-reward trajectories. For each experiment, we tried weight values of (1, 2, 10) by running 1-2 seeds of PEG for each value. We used a weight of 1, 2, 2, 10 for the 4 experiments respectively . PEG uses MPPI, a sample-based optimizer, to optimize the objective. While MPPI has hyperparameters such as the number of samples and number of iteration rounds, these are easy to tune, since optimizer performance increases with the amount of samples and iterations. We therefore just choose as many samples (2000 candidates) and rounds (5 optimization rounds) as we can while keeping training time reasonable.

## F ADDITIONAL EXPERIMENTS

### F.1 PEG WITH OTHER EXPLORATION METHODS

While we chose to train our exploration policy and value function with Plan2Explore, PEG can work with any exploration method as long as it provides an exploration reward and policy. Random Network Distillation (RND) is another powerful exploration method that rewards discovering new

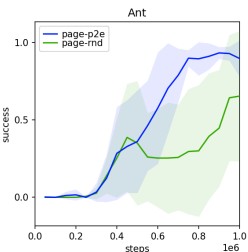

Figure 12: P2E vs. RND

states. We run both variants with 5 seeds in the Ant environment Figure 12. We find that the RND variant still explores the environment, albeit not as well as P2E. We hypothesize P2E exploration is particularly synergistic with our model-based RL framework, since the P2E explorer explicitly seeks transitions $(s, a, s')$ that improve the model. As the model accuracy increases, so does the quality of the goals generated by PEG .

## F.2 MPPI vs CEM for Goal Optimization

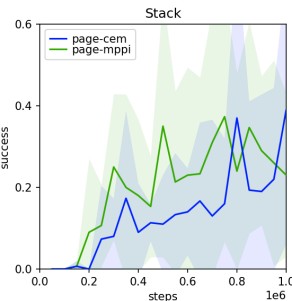

Figure 13: MPPI vs. CEM

We investigate the impact of the goal optimizer on PEG . We choose two popular optimizers, MPPI and CEM, to optimize our objective, and run 5 seeds of each variant in the stacking environment. As seen in Fig. 13, the two agents perform similarly, showing that the PEG objective is robust to the choice of optimizer.

## F.3 Does conditioning on PEG goals maximize exploration?

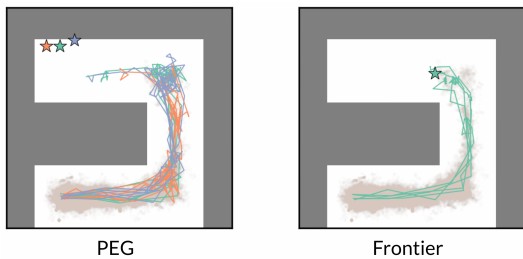

Figure 14: PEG exploration in a U-maze. Brown background dots: explored states, commanded goals: ⋆, resulting paths: colored lines. (Left) PEG optimizes directly for exploration, even setting unseen goals, and achieving farther paths. (Right) Setting goals at the frontier of the seen state distribution yields less exploration.

We assess the importance of accounting for the goal-conditioned policy's capabilities. To do so, we evaluate if states coming from a policy conditioned on PEG goals are more novel than states coming

from the same policy conditioned on other goal strategies. In Figure 14, we take a partially trained agent and its replay buffer, and plot the proposed goals and resulting trajectories for PEG and MEGA (called Frontier in the figure). We find that PEG goals lead the ant to the top left more often. This figure is also used in the intro for exposition.

## F.4    MORE GOALS FOR 3-BLOCK STACKING

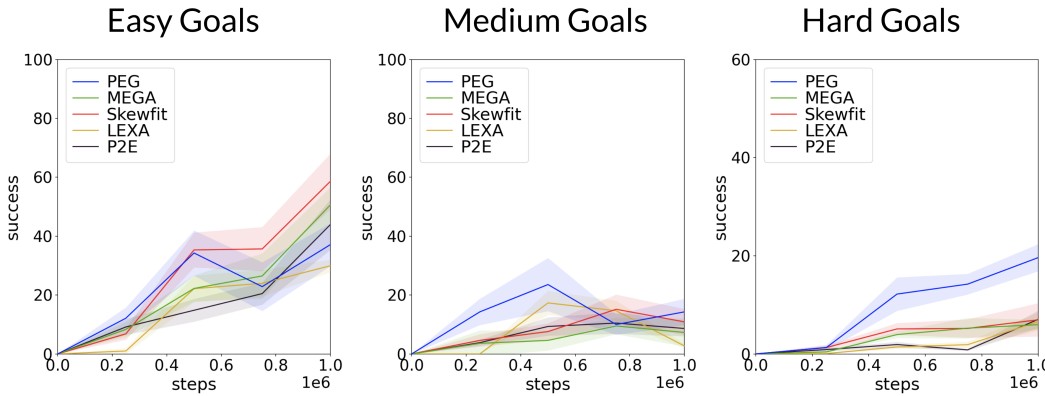

Figure 15: Performance of agents evaluated on different types of goals in the 3-Block environment. All methods are run with 10 seeds. Easy goals require picking up one block, medium goals require stacking two blocks, and hard goals require stacking all 3 blocks.

We have defined three types of goals of increasing difficulty - picking up a single block ("Easy"), stacking two blocks ("Medium"), and stacking three blocks ("Hard"). For each goal type, we set specific evaluation goals by varying the location of the pick / stack as well as the choice and ordering of the blocks. In this way, we create 3 distinct Easy goals, 6 Medium goals, and 6 Hard goals.

First, PEG remains clearly the strongest at 3-block stacking (Hard) as seen in Figure 15. The difficulty-based task grouping permits more interesting analysis: PEG performs competitively with the best approaches on the Easy goals, and its gains over baselines are larger for Medium and Hard goals where exploration is most required. On these harder categories of tasks, PEG is both quickest to achieve non-trivial performance (indicating the onset of exploration) and also achieves the highest peak performance.

An interesting side note here is that better exploration tends to create greater challenges for multi-task goal-conditioned policy function approximation with a fixed capacity policy network. Around the 0.5M steps mark, when PEG starts to achieve Hard goals, it grows marginally weaker on Easy and Medium goals. This kind of "forgetting" is well-studied in the continual learning literature; addressing this well may require expanding policy network capacity over time, as in progressive networks.

