# OpenReview forum: "Planning Goals for Exploration"
_ICLR.cc/2023/Conference — ICLR 2023 notable top 25%_

### Official Review · Reviewer_WR3C · 2022-10-23

**Confidence:** 4
**Correctness:** 3
**Technical Novelty And Significance:** 3
**Empirical Novelty And Significance:** 3
**Recommendation:** 6

**Clarity, Quality, Novelty And Reproducibility:**

Clarity: It is a bit difficult to separate this paper's contribution from existing literature. Specifically, to someone who is not fully aware of LEXA, it is unclear in Section 3 and Algorithm 1 which specific parts are different from LEXA/Go-Explore. Also, there are issues with the details of the method not being described clearly.
Quality: See above weaknesses.
Novelty: The paper mostly combines a lot of existing ideas and methods.

**Strength And Weaknesses:**

Strengths:
- The paper proposes an interesting idea in bringing together Go-Explore and LEXA and proposing a new objective for what goals to choose.
- The visualizations are helpful for building intuition, and the 3-Block Stack results are quite nice. The ablations are also very useful.

Weaknesses:
- It is not clearly described in the main text how the goals in the algorithm are sampled. Maybe I missed this somewhere, but this is a very critical part of the method that is not well described. Can the authors clarify this?
- Relatedly, it seems that the goals chosen by PEG are not in the replay buffer and are in fact sometimes unreachable, e.g. in Figure 5, and this is also mentioned in the paper text. It seems like none of the comparisons make the assumption of sampling from the state space. This is a pretty big difference. Can the authors clarify this?
- All the experiments are using low-dimensional state spaces, although LEXA has been shown to work in settings with high-dimensional inputs. It would be nice to see experiments with vision inputs.



**Summary Of The Paper:**

This paper proposes planning exploratory goals (PEG) for helping an agent explore during training time to achieve diverse goals revealed at test time. PEG brings together the Go-Explore framework, where a goal-conditioned policy brings an agent to a goal, and then an undirected exploration policy explores around that goal, with Latent Explorer Achiever (LEXA), which learns a world model that is used to train an explorer and goal achiever policy.

**Summary Of The Review:**

Due to lack of clarity and issues with the method/evaluation, I think this paper is not ready for acceptance.

Edit: based on additional experiments and clarifications from the authors during the rebuttal period, I have increased my score.

---

> ### Author Response · Authors · 2022-11-12
> **Author Response Part 1/2 (11/11/22)**
>
> Thank you for your review!
> > It is not clearly described in the main text how the goals in the algorithm are sampled. Maybe I missed this somewhere, but this is a very critical part of the method that is not well described. Can the authors clarify this?
>
> Indeed, PEG is about how to select goals to get exploratory behavior during unsupervised Go-Explore exploration, and its core technical aspects are described in Section 3. We apologize for any lack of clarity here, and request the reviewer to please elaborate if particular aspects of the description are confusing.
>
> At a high level, we select goals for the Go-phase that directly maximize the expected exploration value from the future Explore phase (Eq 2). To forecast these *future* exploration rewards, we propose to use a learned world model, eventually approximating Eq 2 as Eq 5. Finally, for the actual optimization, we use an off-the-shelf sampling-based optimizer called MPPI. MPPI starts by evaluating Eq 5 for various randomly sampled goals from the goal space, and then iteratively improves them (through the procedure described in Appendix A.5) to find the argmax. The details of this optimizer are relatively less important, and indeed, in our appendices we report that a different optimizer (CEM) also suffices.
>
> Finally, pertinent to the next response, these final argmax goals are not restricted to lie within the replay buffer or be reachable; rather, we may think of this goal as an arbitrary conditioning vector that is optimized to induce exploratory trajectories when fed into the suboptimal goal-conditioned policy.
>
> > Relatedly, it seems that the goals chosen by PEG are not in the replay buffer and are in fact sometimes unreachable, e.g. in Figure 5, and this is also mentioned in the paper text. It seems like none of the comparisons make the assumption of sampling from the state space. This is a pretty big difference. Can the authors clarify this?
>
> As stated above, the goals selected by PEG are not restricted to lie in the replay buffer or be reachable; rather, we may think of this goal as an arbitrary conditioning vector that is optimized to induce exploratory trajectories when fed into the suboptimal goal-conditioned policy.
>
> Prior goal-sampling-based exploration approaches, such as MEGA and Skewfit, aim to select goals from amongst states that have been reached before, but rarely. Naturally, to optimize such objectives, they must sample goals from the replay buffer. Specifically, MEGA and Skewfit train a density estimator on their replay buffer and choose goals from among low-density states in the replay buffer.
>
> PEG instead optimizes directly for goals that induce exploration. It may be seen as a benefit of this more direct approach that we are not restricted to sample goals from the replay buffer. Instead, PEG’s optimization procedure is free to consider any goal within the full goal space, including goals that lie far outside the observed distribution. MEGA and Skew-fit, which aim to sample low-density states, have no way to meaningfully differentiate between states far from the replay buffer, which all have near-zero density. Note that these out-of-distribution states are not all good goals for exploration; they lie outside the training distribution of the goal-conditioned policy and will in most cases cause haphazard behavior that does not meaningfully approach the commanded goal. As such, these approaches would perform poorly if allowed to sample goals outside the replay buffer.
>
> **To confirm this, we ran an experiment in the Ant Maze with a variant of MEGA that samples from the state space.** We call this variant MEGA-Unrestricted, and plot the mean and standard error of 5 seeds (see Appendix A.3 "MPPI Initial Distribution").
>
> We can see that MEGA-Unrestricted performs much worse than the original MEGA, validating that this method, representative of other methods in this vein including Skewfit, does not permit unrestricted optimization over the goal space, unlike our approach PEG.

---

> > ### Author Response · Authors · 2022-11-12
> > **Author Response Part 2/2 (11/11/22)**
> >
> > > All the experiments are using low-dimensional state spaces, although LEXA has been shown to work in settings with high-dimensional inputs. It would be nice to see experiments with vision inputs.
> >
> > Thank you, we have indeed been striving to increase the dimensionality of the state space where we evaluate our exploration approach. Note however that most prior exploration approaches use environments with low-dimensional state settings where exploration is already very challenging (e.g. MEGA [1], DIAYN [2], DADS [3], GoalGAN [4], Sibling Rivalry [5]). For example, MEGA (2021) evaluated exploration approaches on 2-6 dimensional goal states, our approach evaluates on 2, 9, 14, and 29-dimensional goal states. However, we recognize that larger dimensionality settings such as images may carry even greater challenges and serve as a sterner test for exploration methods. For PEG in particular, our current issue with evaluating the approach on images is computation-related: our block-stacking policies (29-dimensional state) already take 2 days to train, and with larger architectures for visual settings, we expect that training will take even longer. However, we are working on this, and if we are able to make progress within the response period, we will post here.
> >
> > [1] Pitis, Silviu, et al. "Maximum entropy gain exploration for long horizon multi-goal reinforcement learning." International Conference on Machine Learning. PMLR, 2020.
> >
> > [2] Eysenbach, Benjamin, et al. "Diversity is all you need: Learning skills without a reward function." arXiv preprint arXiv:1802.06070 (2018).
> >
> > [3] Sharma, Archit, et al. "Dynamics-aware unsupervised discovery of skills." arXiv preprint arXiv:1907.01657 (2019).
> >
> > [4] Florensa, Carlos, et al. "Automatic goal generation for reinforcement learning agents." International conference on machine learning. PMLR, 2018.
> >
> > [5] Trott, Alexander, et al. "Keeping your distance: Solving sparse reward tasks using self-balancing shaped rewards." Advances in Neural Information Processing Systems 32 (2019).
> >
> > > Clarity: It is a bit difficult to separate this paper's contribution from existing literature. Specifically, to someone who is not fully aware of LEXA, it is unclear in Section 3 and Algorithm 1 which specific parts are different from LEXA/Go-Explore. Also, there are issues with the details of the method not being described clearly.
> >
> > > Novelty: The paper mostly combines a lot of existing ideas and methods.
> >
> > Thank you for raising these concerns. We believe they were caused largely by deficiencies in presentation, and have now improved those aspects significantly. Please see the global post (not addressed to any specific reviewer), where we more clearly explain the connection to LEXA / Go-Explore. Also, please let us know if our summary of PEG above helped clear up any confusion over the method details, or whether there is anything else we could elaborate on more clearly.

---

> > > ### Author Response · Authors · 2022-11-17
> > > **Reviewer WR3C Followup (11/16/22)**
> > >
> > > Dear reviewer WR3C, thank you again for your feedback. As the open discussion period draws to a close, we wanted to check back to see whether you have any remaining concerns. We believe that we have sufficiently responded to your earlier queries on various aspects of this work and its connection to prior work, but would be happy to clarify further, and grateful for any other feedback.

---

> > > > ### Comment · Reviewer_WR3C · 2022-11-17
> > > > **Additional clarifications are needed**
> > > >
> > > > Thanks to the authors for the response. I would like to request some additional clarifications:
> > > >
> > > > > ... these final argmax goals are not restricted to lie within the replay buffer or be reachable; rather, we may think of this goal as an arbitrary conditioning vector...
> > > >
> > > > How is this conditioning vector chosen at implementation time, and how is the goal space for a problem setting chosen? And more importantly, it does not seem like a benefit that the goals are not restricted to lie within the replay buffer--in particular, this requires an additional assumption: PEG assumes the ability to sample from the full goal space instead of just the replay buffer.
> > > > The reason why I think this is a drawback is because it does not very scalable, especially to higher dimensions. I'm confused on how this method even be appiled with image observations, where the goal space is images and therefore can't be easily sampled from. Can the authors clarify? It is nice to hear that the authors are working on a setting with image observations although bottlenecked by training being compute-heavy, and it would be good to see the results and details from these experiments.

---

> > > > > ### Author Response · Authors · 2022-11-18
> > > > > **Reviewer WR3C Followup Part 1/2 (11/17/22)**
> > > > >
> > > > > Thank you for following up! Your concerns are now much clearer to us, and we better address them below.
> > > > >
> > > > > ### How the MPPI optimization within PEG works
> > > > >
> > > > > The conditioning goal vector is the output of the MPPI optimizer that optimizes the PEG objective of Eq 5. MPPI is an iterative optimization method that starts from some initial goal candidates, evaluates the objective for each, then fits a distribution to the most promising candidates, then resamples new candidates from this distribution, and repeats. A couple of notes:
> > > > >
> > > > > A. When the optimizer works well, it should be relatively insensitive to the choice of initial goal candidates (as long as they are quite diverse), and
> > > > >
> > > > > B. No explicit constraints are imposed on the final output of MPPI to be from the same distribution as the initial goal candidates.
> > > > >
> > > > > ### What is the goal space in which MPPI optimization operates, and how is it chosen?
> > > > >
> > > > > Throughout the paper, we assume a very general goal space, which is identical to the state space ($\mathcal{G} = \mathcal{S}$), as stated in Section 2 in the paper. In other words, any state is a plausible goal, and nothing is handcrafted for each environment.
> > > > >
> > > > > Sidenote: This choice of goal space G avoids replicating assumptions from some prior works. For example, the original experiments in MEGA [1] restrict goals to only those dimensions of the state space which are relevant to the test tasks (i.e., $\mathcal{G} \subset \mathcal{S}$). Consequently, for AntMaze, they set the goals to the XY position of the center of mass of the ant, while the full state space is 28 dimensions. This effectively leaks some information about the test tasks into the exploration procedure, reducing the unsupervised exploration challenge significantly, and training a more limited goal-conditioned task policy that can only reach XY goals.
> > > > >
> > > > > ### "PEG-Restricted’’ with initial goal candidates sampled from the replay buffer?
> > > > > Throughout our paper, we initialize the MPPI optimizer with goal vectors sampled uniformly at random from the full goal space. One alternative would have been to restrict the initial candidates to lie in the agent’s replay buffer, but we found in practice that this restriction made no difference and it was more convenient for implementation reasons to simply sample uniformly at random from the unrestricted goal space (same as the state space), e.g.. using the joint limits of the various robot joints. In Appendix A.3 ``MPPI Initial distribution’’, we now show results of running PEG with MPPI initialization from the replay buffer (“PEG-Restricted”), in the AntMaze environment. Its performance is virtually indistinguishable from the original PEG method. This is to be expected, and is a sign that the optimizer is working well (see note A above). We have updated the MPPI implementation details in appendix A.3 with this discussion and experiment.
> > > > >
> > > > >
> > > > > ### Could other methods benefit from sampling outside the replay buffer?
> > > > > Recall further that in an earlier experiment (see appendix A.3 section ``MPPI initial distribution”), we allowed our strongest baseline MEGA to also sample goals from outside the replay buffer, i.e., the full goal space (now highlighted in bold in our [post from 11/16](https://openreview.net/forum?id=6qeBuZSo7Pr&noteId=IAyxls_F1G)). However, these methods rely on heuristic objectives for selecting goals (like “low density” of the visited state distribution) which only approximate the correct goal-setting objective for states close to the replay buffer. Therefore, “MEGA-unrestricted” is unable to sample effective goals outside the replay buffer, and fares very poorly. In contrast, PEG directly optimizes the exploration value of goal states, and permits choosing meaningful goals even in this unrestricted setting. This is an advantage, since it permits optimizing over a larger space of goal candidates, which leads to better goals.
> > > > >
> > > > > ### Is sampling from the unrestricted goal space scalable?
> > > > > As noted above in note B, even when PEG is initialized with replay buffer states as in PEG-Restricted above, MPPI assumes the ability to sample other goals in the goal space, and indeed its final output may very well lie outside the replay buffer. For image-based state spaces, a naive application would involve sampling random image goals, which is unlikely to scale, as the reviewer correctly notes. However, rather than directly applying MPPI-style iterative search in such high-dimensional settings, it is common practice to train a compressed latent representation such as through a variational autoencoder (VAE), in which optimization becomes much more tractable  [2]. This is the approach we are currently attempting for PEG in image-based environments.

---

> > > > > > ### Author Response · Authors · 2022-11-18
> > > > > > **Reviewer WR3C Followup Part 2/2 (11/17/22)**
> > > > > >
> > > > > > ### Summary:
> > > > > > By directly optimizing for exploration value when selecting goals, PEG is uniquely able to select goals from the full goal space even beyond the replay buffer, where competing methods fail. The goal space in all our experiments is simply the full state space. For higher-dimensional settings like images, we believe based on prior work that sampling and optimizing over goals that are not in the replay buffer is still tractable, through learning a compressed goal representation.
> > > > > >
> > > > > > Thank you for engaging with us again during this response period and allowing us the opportunity to better address your concerns. Please do not hesitate to request any further clarifications if needed.
> > > > > >
> > > > > > [1] Pitis, Silviu, et al. "Maximum entropy gain exploration for long horizon multi-goal reinforcement learning." International Conference on Machine Learning. PMLR, 2020.
> > > > > >
> > > > > > [2] Nair, Ashvin V., et al. "Visual reinforcement learning with imagined goals." Advances in neural information processing systems 31 (2018).

---

> > > > > > > ### Comment · Reviewer_WR3C · 2022-11-18
> > > > > > > **Response to clarifications**
> > > > > > >
> > > > > > > Thanks to the authors for the many clarifications in the follow-up response, which have better addressed my original concerns. It would make the paper and method a lot more convincing to see experiments using image observations and learning a compressed goal representation, as alluded to by the authors above, as well as having a better discussion of such limitations, and I'd recommend having these for the next version of the paper. I appreciate the authors' responses during the rebuttal period and will update my score to reflect my updated views on the paper.

---

> > > > > > > > ### Author Response · Authors · 2022-11-18
> > > > > > > > **Thank you!**
> > > > > > > >
> > > > > > > > We are happy to hear that we have better addressed your concerns and will incorporate the feedback and discussion into the next version of the paper.

---

### Official Review · Reviewer_p4aC · 2022-10-24

**Confidence:** 5
**Correctness:** 4
**Technical Novelty And Significance:** 2
**Empirical Novelty And Significance:** 3
**Recommendation:** 8

**Clarity, Quality, Novelty And Reproducibility:**

As stated above in the review, the clarity, quality and reproducibility aspects are mostly on point. The reproducibility comes with a relatvie caveat, but remains reasonable regarding the standards in the field.

I firmly believe the reproducibility aspect could be reasonably improved through the authors/reviewers discussions and am looking forward to it regarding the parameter tuning, and resource usage differential between the proposed conceptual methods and the benchmarked alternatives from the state-of-the-art. The amount of focus to explain the differential between the proposed contribution, the original framework on which they build upon, would be more challenging, as it implies a substantial rewriting part.


**Details Of Ethics Concerns:**

None applicable.

**Strength And Weaknesses:**

# Strengths:

- (1) Readability.

Overall, the readability is fair. The main ideas are well conveyed, justified and articulated while mostly providing sufficient context to the reader, albeit sometimes superficially (cf. below for the downsides). The grammar, and overall the writing, is relatively mature in its current state.

- (2) Experimental Evaluation.

This one is a blend of benchmark per se, engineering to re-implement pertinent baselines and ablative study of the proposed conceptual contribution.

- (3) The Performance of the proposed method.

- (4) The Reproducibility.

This one is somewhat of a mixed bag. The positive aspects predominently lie in the proactive care the authors have put into explaining the experimental set-up, the re-implementation of the baselines and the description of the considered dataset environments.
In absence of an existing, public, standardized and unified benchmark environment, the proposed package is fair on many aspects (a few hicups though, as discussed below).

# Weaknesses:
Despite several positive aspects to the submitted paper, there are several noticeable shortcomings. In particular,


- (1) How much does it cost?

While the experimental evaluation which focuses on the task-oriented performance per se, alternative considerations could be discussed to help better understand the relative positioning of the proposed method and how it contrasts (favourably or not) on algorithmic and resource usage and management considerations.

- (2) Related Work.
The section is awkwardly placed as a #5 section, right before the conclusion and is relatively short.
It is rather superficial by often characterizing methods at a very top-level, rather than focusing on their key differanciations and relative interconnections from a mechanistic-centric point of view.

To a lesser extent, it also ommits quite a few pertinent work in the field, e.g.,

(Sub-goal oriented, and references therein)
* CHANE-SANE, Elliot, SCHMID, Cordelia, et LAPTEV, Ivan. Goal-conditioned reinforcement learning with imagined subgoals. In : International Conference on Machine Learning. PMLR, 2021. p. 1430-1440.


(Seminal formulation of several related concepts)
* YAMAUCHI, Brian. Frontier-based exploration using multiple robots. In : Proceedings of the second international conference on Autonomous agents. 1998. p. 47-53.

* YAMAUCHI, Brian. A frontier-based approach for autonomous exploration. In : Proceedings 1997 IEEE International Symposium on Computational Intelligence in Robotics and Automation CIRA'97.'Towards New Computational Principles for Robotics and Automation'. IEEE, 1997. p. 146-151.

- (3) The emphasis should be much stronger to explain the difference between the added contritbution and the vanilla prior work from Mendonca NeurIPS 2021.

The corresponding section (slightly more than a 1-pager) to describe it, including the original (prior work) work on which this one is based is way too shallow to support the current state of the submissions.

- (4) Reproducibility:

Additionally, despite having many implementation details provided, the parameter tuning seems relatively simplistic and unchallenged.


**Summary Of The Paper:**

The paper proposes a novel Reinforcement Learning based unsupervised method for the task of goal planning for agent based exploration in a non task-centric paradigm.

**Summary Of The Review:**


As it currently stands, the paper has several positive aspects rooting for it. Not having access to the code and finer graine details
regarding the intrinsics and experiments is a slight turn-off, but regarding the standards of the field and the venue, it remains on the acceptable side.

The downsides however, include a lack of experimental challenge of the parameter tuning and the overal descriptive focus is not emphasized enough on the conceptual differential between the proposed method and the original prior work it builds upon, i.e., Mendonca et al. NeurIPS 2021.

Overall, it is hence slightly bellow borderline but could I would be happy to revise my stance and discuss the (still/currently) missing bits.

---

> ### Author Response · Authors · 2022-11-12
> **Author Response Part 1/2 (11/11/22)**
>
> Thank you for your review!
> > (1) How much does it cost?
> While the experimental evaluation which focuses on the task-oriented performance per se, alternative considerations could be discussed to help better understand the relative positioning of the proposed method and how it contrasts (favourably or not) on algorithmic and resource usage and management considerations.
>
> > (4) Reproducibility:
> Additionally, despite having many implementation details provided, the parameter tuning seems relatively simplistic and unchallenged.
>
> >I firmly believe the reproducibility aspect could be reasonably improved through the authors/reviewers discussions and am looking forward to it regarding the parameter tuning, and resource usage differential between the proposed conceptual methods and the benchmarked alternatives from the state-of-the-art.
>
> Thank you for this question. We have tried to consider key relevant costs below, but please let us know if there are others we should report.
>
> We have updated the implementation details section and appendix with more information on runtime and compute resources, and summarize below for convenience. Note that we do not claim to be faster than baselines in walltime or resources, rather that any computational overheads introduced by our approach are minimal compared to the rest of the MBRL pipeline.
>
> Specifically, the runtime and  resource usage between methods did not differ significantly, as everything is implemented on top of the backbone LEXA model-based RL agent. Runtime is dominated by neural network updates of the policies and world model, not the goal selection routine defined by each method. Each seed was run on 1 GPU (Nvidia 2080ti or Nvidia 3090) and 4 CPUs, and required ~11GB of GPU memory.
>
> Total training time by experiment (roughly same for all methods)
> |  | Total Runtime (Hours) | Episodes | Episode Length | Seconds per Episode |
> |:---:|:---:|:---:|:---:|:---:|
> | Point Maze | 16 | 12000  | 50 | 4.8 |
> | Walker | 16 | 5000  | 150 | 11.52 |
> | AntMaze | 48 | 2000  | 500 | 86.4 |
> | 3-Block Stack | 48 | 6666 | 150 | 25.9 |
>
> We benchmarked the goal selection procedure for PEG and the other goal selection baselines in the block stacking environment and recorded the average wall clock time.
>
> |  | Seconds / Episode |
> |:---:|:---:|
> | PEG: | 0.51 |
> | MEGA: | 0.48 |
> | SkewFit: | 0.46 |
>
> We can see that there is little difference in speed between methods, and the overhead introduced by PEG goal selection is minimal to LEXA, and competitive with other goal selection baselines. Because LEXA does not select goals, it finishes up to an hour earlier than goal setting methods in our experiments. As a sidenote, the times reported above are amortized over episodes, since we compute a set of 50 goals every 50 episodes for all methods, rather than computing 1 goal per episode.
>
>
> For parameter tuning, we used the default hyperparameters of the backbone MBRL agent (e.g. learning rate, MLP size, etc.) and kept them constant across all baselines. PEG only has 1 hyperparameter that requires tuning - the reward weight, which affects the MPPI update distribution. The higher the weight, the more we update the goal sampling distribution towards high-reward trajectories. For each experiment, we tried weight values of  (1, 2, 10) by running 1-2 seeds of PEG for each value. We used a weight of 1, 2, 2, 10 for the 4 experiments respectively.
>
> PEG uses MPPI, a sample-based optimizer, to optimize the objective. While MPPI has hyperparameters such as the number of samples and number of iteration rounds, these are easy to tune, since optimizer performance increases with the amount of samples and iterations. We therefore just choose as many samples (2000 candidates) and rounds (5 optimization rounds) as we can while keeping training time reasonable.

---

> > ### Author Response · Authors · 2022-11-12
> > **Author Response Part 2/2 (11/11/22)**
> >
> > > (2) Related Work. The section is awkwardly placed as a #5 section, right before the conclusion and is relatively short. It is rather superficial by often characterizing methods at a very top-level, rather than focusing on their key differanciations and relative interconnections from a mechanistic-centric point of view.
> >
> > > (3) The emphasis should be much stronger to explain the difference between the added contritbution and the vanilla prior work from Mendonca NeurIPS 2021.
> > The corresponding section (slightly more than a 1-pager) to describe it, including the original (prior work) work on which this one is based is way too shallow to support the current state of the submissions.
> >
> > > The downsides however, … the overal descriptive focus is not emphasized enough on the conceptual differential between the proposed method and the original prior work it builds upon, i.e., Mendonca et al. NeurIPS 2021.
> >
> > Thank you for raising these concerns. We believe they were caused largely by deficiencies in presentation, and have now improved those aspects significantly. Please see the global post (not addressed to any specific reviewer), where we more clearly discuss background and related work and differentiate PEG from them.
> >
> >
> > > To a lesser extent, it also ommits quite a few pertinent work in the field, e.g.,
> > (Sub-goal oriented, and references therein) ...
> >
> > Thank you for these references, we have now referenced frontier exploration in section 3, and  updated related work to discuss works that improve goal-conditioned policy training like Chane-Sane et. al.  We summarize the discussion below:
> >
> > PEG falls under the family of goal-directed exploration methods, which aim to propose goals for a GCRL agent for exploration. Related to goal-directed exploration are methods that focus on training goal-conditioned policies. Andrychowiscz et al. proposes an efficient goal relabeling strategy, and Chane-Sane uses imagined subgoals to guide the policy search process.
> >
> > > Not having access to the code and finer graine details regarding the intrinsics and experiments is a slight turn-off, but regarding the standards of the field and the venue, it remains on the acceptable side.
> >
> > Please note that we submitted the PEG codebase at time of original submission as part of the supplemental file. It includes our approach, all baseline implementations, and hyperparameters. We plan to expand this further to include environment setup details, upon acceptance.

---

> > > ### Author Response · Authors · 2022-11-17
> > > **Reviewer p4aC Followup (11/16/22)**
> > >
> > > Dear reviewer p4aC, thank you again for your feedback. As the open discussion period draws to a close, we wanted to check back to see whether you have any remaining concerns. Your review contained two key concerns: PEG’s resource usage and algorithmic costs compared to baselines, and also PEG’s relationship to prior work. We believe that we have addressed both concerns now, but would be grateful for any other feedback.

---

> > > > ### Comment · Reviewer_p4aC · 2022-11-19
> > > > **Reply and rating upgrade**
> > > >
> > > > Dear authors,
> > > >
> > > > I consider that most of the substantial concerns have been thoroughly addressed during this discussion phase; not only regarding the aspects I had initially raised, but throughout the reviews. As such, I am happy to bump my rating but, quite obviously, can only stress the importance to include as much of the updated contents to the cam-ready version of the paper to make it a substantial improvement with regards to the qulity standards of the venue.

---

### Official Review · Reviewer_i4Jc · 2022-10-24

**Confidence:** 4
**Correctness:** 4
**Technical Novelty And Significance:** 3
**Empirical Novelty And Significance:** 3
**Recommendation:** 8

**Clarity, Quality, Novelty And Reproducibility:**

**Very very minor issues (which have *NOT* been considered for decision making)**

6. In Appendices A.4 and A.5 PAGE is used instead of PEG. I believe in an earlier version of the manuscript, the method was named PAGE.

7. Typo: Page 8 last line "achievd" --> "achieved"

8. Please slow down the GIFs speed on the webpage. It is incredibly hard to view the visualizations.


**Reproducibility**
`
I appreciate the authors open-sourcing the code in the supplementary material.


**Strength And Weaknesses:**

**Strengths**

1. The idea is novel, and the paper is clearly written, making it easy for the reader to understand the motivation as well as the method.

2. Thorough experiments to validate the claim of optimizing goal commands directly.


**Weaknesses**

**Claims (clarification)**

3. The authors distinguish directly optimizing for goal commands with exploration methods that aim to reach novel states, and say:

    > Note that this does not mean merely commanding the agent to novel or rarely observed states. Instead, PEG commands might be to a previously observed state... PEG only cares that the command will induce the chained GCRL and exploration phases together to generate interesting training trajectories, valuable for policy improvement.

    Shouldn't it be the case that the "interesting trajectory" would encounter some novel states -- otherwise the total exploration reward for that trajectory wouldn't be maximum?


**Methodology**

4. From equations (2) and (5) and the choice of K=1, is it correct to say the following:

    $\max_g \mathbb{E}_{p_\pi G(.|., g)}(s_T)[V^E(s_T)] ≈ \frac{1}{K} \sum_k V_\theta ^E(s_T^k) = V_\theta ^E(s_T)$



    a) What could be the reason that K>1 doesn't have a large effect on the performance?

    b) Additionally, in appendix A.5, where CEM is compared with MPPI, what were the hyperparameters for CEM? How does CEM perform with more number of trajectories?


**Citations**

5. In the Conclusion section, please also cite [1] along with [2] and [3]. Works [1] and [2] were independently published around the same time and share the same idea.

------

**References**

[1] Alexander Neitz, Giambattista Parascandolo, Stefan Bauer, Bernhard Schölkopf, Adaptive Skip Intervals: Temporal Abstraction for Recurrent Dynamical Models, In NeurIPS 2020.

[2] Dinesh Jayaraman, Frederik Ebert, Alexei A Efros, and Sergey Levine. Time-agnostic prediction:Predicting predictable video frames. ICLR, 2019.

[3] Karl Pertsch, Oleh Rybkin, Frederik Ebert, Chelsea Finn, Dinesh Jayaraman, and Sergey Levine. Long-horizon visual planning with goal-conditioned hierarchical predictors. In NeurIPS 2020.

**Summary Of The Paper:**

Existing exploration methods tend to explore at the frontier of the currently seen states. The authors of this paper propose to direct the goal-conditioned policy in a way that induces higher exploration value trajectories for the agent. Specifically they first use the goal-conditioned policy $\pi^G$ to reach a such a state $s_T$ that maximizes the exploration value $V^E(s_T)$.

Extensive experiments on control as well as robotic manipulation showcase the sample-efficiency of the method. In particular, the method achieves really good results on hard task of block-stacking which the previous works fail at.

**Summary Of The Review:**

Given the clear motivation, methodology and experimentation of the paper, I lean towards accepting the paper (*6: marginally above the acceptance threshold*). However, I'd want the authors to address my concerns as well as other reviewers' comments during the rebuttal phase, and my final decision would be subject to the rebuttal.

-----

Post-rebuttal update:

I've updated my score to 8 -- good paper, accept after going through other reviewers' comments and rebuttal. The authors' rebuttal addresses my concerns.

---

> ### Author Response · Authors · 2022-11-12
> **Author Response (11/11/22)**
>
> Thank you for your review!
> > The authors distinguish directly optimizing for goal commands with exploration methods that aim to reach novel states, and say:
> Note that this does not mean merely commanding the agent to novel or rarely observed states. Instead, PEG commands might be to a previously observed state... PEG only cares that the command will induce the chained GCRL and exploration phases together to generate interesting training trajectories, valuable for policy improvement.
>
> > Shouldn't it be the case that the "interesting trajectory" would encounter some novel states -- otherwise the total exploration reward for that trajectory wouldn't be maximum?
>
> Yes, interesting trajectories should encounter novel states, however a key idea in our approach is that commanding the goal-conditioned policy (GCP) to novel states is not the best way to encounter novel states. Instead, we must account for suboptimalities in the GCP (as we argue at the bottom of Pg 1). Thus, it is very possible that conditioning the GCP on a previously seen goal, or even a completely nonsensical goal (e.g. XY locations outside the bounds that the environment sets) could result in the optimal exploratory behavior.
>
> > From equations (2) and (5) and the choice of K=1, is it correct to say the following…
>
> Yes, the K=1 case is correct.
> > a) What could be the reason that K>1 doesn't have a large effect on the performance?
>
> This choice was made through empirical experimentation: in our initial runs with K > 1, we found that it provided little gain over K=1 and at the cost of increased computation time. In our settings, just collecting K=1 trajectory from the goal-conditioned policy was enough to sufficiently represent the terminal state distribution, because this distribution is very narrow: our environments (following prior works like MEGA, LEXA, Skewfit) have deterministic dynamics, and randomness arises solely from the stochasticity of the goal-conditioned policy $\pi^G$. Empirically, we find $\pi^G$ to be only minimally stochastic during training.
>
> > b) Additionally, in appendix A.5, where CEM is compared with MPPI, what were the hyperparameters for CEM? How does CEM perform with more number of trajectories?
>
> In the CEM vs MPPI experiment, we used the following hyperparameters for the optimizers. Note that CEM and MPPI follow a similar iterative optimization procedure, with the only difference arising in the refitting of the sample distribution. CEM uses the Top K trajectories to compute the new sampling distribution, whereas MPPI uses all trajectories, weighted by their cost, to compute the new sampling distribution.
>
> Shared Hyperparameters for CEM / MPPI:
> - Number of trajectories: 2000
> - Trajectory length: 250
> - Optimization rounds: 5
>
> CEM specific hyperparameters:
> - Top K: 20%
>
> MPPI specific hyperparameters:
> - Cost scale: 1.0
>
> We find that both MPPI and CEM improves with more trajectories, at the cost of compute time.
>
> > In the Conclusion section, please also cite ...
> > Very very minor issues...
>
> Thank you for pointing out relevant work and typos, we have addressed these in the latest version of the website and draft, with blue text to highlight the changes.

---

> > ### Author Response · Authors · 2022-11-17
> > **Reviewer i4Jc Followup (11/16/22)**
> >
> > Dear reviewer i4Jc, thank you again for your feedback. As the open discussion period draws to a close, we wanted to check back to see whether you have any remaining concerns. We believe that we have sufficiently responded to your earlier queries on various aspects of this work, but would be happy to clarify further, and grateful for any other feedback.

---

> > > ### Comment · Reviewer_i4Jc · 2022-11-22
> > > **Thanks for the extensive rebuttal experiments + Final Decision**
> > >
> > > Thanks to the authors for their rebuttal.
> > >
> > > I think the reasoning about $K=1$ working seems reasonable to me.
> > > > because this distribution is very narrow: our environments (following prior works like MEGA, LEXA, Skewfit) have deterministic dynamics, and randomness arises solely from the stochasticity of the goal-conditioned policy . Empirically, we find  to be only minimally stochastic during training.
> > >
> > > I agree that typical deepmind control environments and the LEXA environments are very minimally stochastic and don't really have an added advantage of multiple $K$!
> > >
> > > Having gone through the rebuttal from authors I vote for an accept of the paper.

---

### Official Review · Reviewer_rUvT · 2022-10-25

**Confidence:** 4
**Correctness:** 3
**Technical Novelty And Significance:** 3
**Empirical Novelty And Significance:** 2
**Recommendation:** 8

**Clarity, Quality, Novelty And Reproducibility:**

The paper is written with clear motivations. The approach and experimental sessions are easy to follow. The method, while simple, is effective.

**Strength And Weaknesses:**

The authors contributed a useful method that speeds up exploration thus resulting in less self-exploration time for agents in a new environment to achieve goals in the furthest states. Their experiment environments are diverse and the results are strong.

However, the test goals are selected from a specific distribution of states that require the furthest explorations. In Figure 4, the variance for PEG is very high, which makes the results less convincing given only five seeds. What are the authors' thoughts on an experimental design that could reduce the variance?

Furthermore, the reviewer would like the authors to address the following questions to make the conclusions more convincing:

1. About Algorithm 1, since training prioritizes exploration, how much is exploitation performance affected (e.g. the number of steps to achieve a goal sampled uniform at random in the environment) compared to other baselines?
2. in Figure 6, many goals are sampled close to the start state (red dots on the left of the blue dot) and a few around the first corner, which increases more in quantity than those in remote states throughout training. Why might that be the case?

**Summary Of The Paper:**

The authors proposed a method to quickly explore a state space and achieve goals that require long-horizon trajectories. They define the exploration value of a trajectory based on learned transition functions and policies for sampling trajectories, which are added to the training buffer. Their experiments on four tasks from 2D point navigation to 3D locomotion and 3-block stacking demonstrate the effectiveness of their method in comparison to two other baselines for long-horizon multi-goal RL.

**Summary Of The Review:**

This is a well-written paper on an important topic. The method has the potential to benefit robot navigation and manipulation in new environments. While the reviewer has concerns over the variance of one of their experiments, the results are overall very convincing.

---

> ### Author Response · Authors · 2022-11-12
> **Author Response Part 1/2 (11/11/22)**
>
> Thank you for your thoughtful review!
> > In Figure 4, the variance for PEG is very high, which makes the results less convincing given only five seeds. What are the authors' thoughts on an experimental design that could reduce the variance?
>
> The 3-block stacking experiment is a hard exploration task, and we notice that runs are bimodal. They either succeed, or fail completely to get off the ground. Out of 5 runs, PEG succeeded in 4 and failed completely in 1, hence the high variance. Baselines also exhibited high variance where most seeds were either completely flat or achieved very low success rates under 10%.
>
> We will seek to increase confidence in those gains through retraining PEG and baselines with more random seeds. These are computationally intensive experiments, but we will report our progress before the end of the response period.
>
> > However, the test goals are selected from a specific distribution of states that require the furthest explorations.
>
> On the block stacking environment, rather than only evaluating methods on the very hardest goal of 3-block stacking, we have now compared all methods on more tasks. We have defined three types of goals of increasing difficulty - picking up a single block, stacking two blocks, and then stacking three blocks. For each goal type, we set specific evaluation goals by varying the location of the pick / stack as well as the choice and ordering of the blocks. In this way, we create 3 distinct “picking” goals, 6 “2-block stacking” goals, and 6 “3-block stacking” goals (expanding from the lone 3-block stacking goal we had reported in the submission). We evaluate PEG and baselines (after 1M training steps) on these goals, and report the mean success rates and standard errors over 5 seeds below:
> |  | Picking | 2-Stack | 3-Stack |
> |:---:|:---:|:---:|:---:|
> | PEG | 42.66% ± 10.97 | 17.66% ± 4.43 | 19.64% ± 2.75 |
> | MEGA | 50.65% ± 5.92 | 7.30% ± 4.75 |  5.97% ± 1.00 |
> | Skewfit | 58.65% ± 9.40 | 10.90 ± 4.40 | 6.96% ± 3.40 |
> | P2E |  43.96% ± 8.56 | 8.65% ± 2.64 | 7.06% ± 1.78 |
> | LEXA | 29.96% ± 2.10 | 2.73% ± 0.71 |  6.59% ± 1.82 |
>
> These results shed clearer light on our original block stacking results. Even though the baselines fail completely on the single 3-block stacking task reported in the submission, they do achieve non-trivial exploration behavior: this is clear from the fact that they can train policies for easier goals such as picking, and to some extent, 2-block stacking. Even on the expanded set of six 3-block stack goals, baseline performance remains poor (highest 7% compared to PEG 20%). PEG performs the best on both 2-stack and 3-stacks, showing that it explores these harder-to-reach stacking states more than the baselines.
>
> As a side note, we observe that the poorer exploration approaches fare marginally better on the easiest task, single-block picking. We believe that this is because for all methods, policies are trained on goal states sampled from the replay buffer (Step 7 of Algorithm 1). PEG explores more diverse states, thus the PEG goal-conditioned policy is required to train on many more diverse goals than MEGA or Skewfit, and policy optimization on this larger task space is more difficult. Instead, the MEGA or Skewfit policies can specialize very well to block-picking, ignoring the other harder tasks that are less represented in their replay buffers.

---

> > ### Author Response · Authors · 2022-11-12
> > **Author Response Part 2/2 (11/11/22)**
> >
> > > About Algorithm 1, since training prioritizes exploration, how much is exploitation performance affected (e.g. the number of steps to achieve a goal sampled uniform at random in the environment) compared to other baselines?
> >
> > Note that PEG goals are only used to command the agent to generate data for the replay buffer, and are not used to train the goal-conditioned policy. For training the goal-conditioned policy, PEG (and all other methods) use the same procedure: train the policy in the world model, where it is trained to achieve goals sampled from the replay buffer. Therefore, as long as trajectories generated by running PEG cover the test time goal distribution, we can expect good exploitation performance.
> >
> > Here, we conducted a follow-up experiment by evaluating uniform goals in the AntMaze environment, as seen below. We divide the maze up into 8 regions, where the regions increase in distance from the start. Each region contains 40 goals that differ in ant orientation.  We record the average success rate and standard error of 5 seeds, for each region.
> >
> > |  | Region 1 | Region 2 | Region 3 | Region 4 | Region 5 | Region 6 | Region 7  | Region 8 |
> > |:---:|:---:|:---:|:---:|:---:|:---:|:---:|:---:|:---:|
> > | PEG | 100% ± 0 | 99.00% ± 0.55 | 98.50% ± 0.89 | 95.00% ± 2.11 | 90.00% ± 4.18 | 86.00% ± 4.39 | 95.00% ± 1.41 | 86.00% ± 5.85 |
> > | MEGA | 100% ± 0 | 100% ± 0 | 100% ± 0 | 96.00% ± 1.14 | 98.00% ± 0.83 | 95.50% ± 0.44 | 95.00% ± 1.22 | 73.50% ± 14.36 |
> > | Skewfit | 100% ± 0 | 99.00% ± 0.55 | 99.50% ± 0.44 | 80.00% ± 8.45 | 84.50% ± 4.49 | 82.00% ± 7.49 | 85.98% ± 6.57 | 69.00% ± 8.2 |
> > | P2E | 100% ± 0 | 100% ± 0 | 99.50% ± 0.44 | 91.98% ± 4.08 | 95.00% ± 2.34 | 94.50% ± 1.92 | 88.00% ± 4.14 | 80.98% ± 8.17 |
> > | LEXA | 100% ± 0 | 100% ± 0 | 98.33% ± 0.52 | 53.33% ± 14.84 | 61.67% ± 7.09 | 4.17% ± 1.39 | 0% ± 0 | 0% ± 0 |
> >
> > The new block stacking experiment and Ant Maze experiment shows that PEG does well on all types of goals, particularly in the harder-to-explore goals, while baselines start degrading as the goal difficulty increases. Note that by evaluating at the end of training, most of the baselines do well in all regions except for the last region. We plan to evaluate these goals during the course of training to more clearly see when different methods explore different parts of the maze.
> >
> > > in Figure 6, many goals are sampled close to the start state (red dots on the left of the blue dot) and a few around the first corner, which increases more in quantity than those in remote states throughout training. Why might that be the case?
> >
> > Figure 6 projects the 29 dimensional state of the ant, down to 2-D spatial location of the center of mass. The other dimensions contain the Ant’s joint positions and velocities. Therefore, the goal states that are close to the sample states may contain interesting values in the other dimensions, such as proposing unusual configurations of the Ant like flipping it upside down or hovering in the air. On the [website](https://sites.google.com/view/exploratory-goals/home#h.tfthqa195kf), we randomly sample 10 PEG goals near the start state, and see that they actually contain exploratory, hard-to-reach body configurations.

---

> > > ### Author Response · Authors · 2022-11-17
> > > **Reviewer rUvT Followup (11/16/22)**
> > >
> > > Dear reviewer rUvT, thank you again for your feedback.  Per your comments (and those of reviewer 5sEN) regarding more evaluations, we have now posted a global response with new results that point more clearly to PEG’s gains over the baselines.
> > >
> > > Separately, as the open discussion period draws to a close, we wanted to check back to see whether you have any remaining concerns. Your review pointed to two key concerns: the variance of the block stacking results, and possible bias in the selection of test time goals. We believe that we have addressed both now, but would be grateful for any further feedback.

---

> > > > ### Comment · Reviewer_rUvT · 2022-11-21
> > > > **I have no more concerns**
> > > >
> > > > Thank you for conducting the follow-up experiments. I have no more concerns about the paper being strongly accepted.

---

### Official Review · Reviewer_5sEN · 2022-10-25

**Confidence:** 3
**Correctness:** 3
**Technical Novelty And Significance:** 3
**Empirical Novelty And Significance:** 3
**Recommendation:** 8

**Clarity, Quality, Novelty And Reproducibility:**

The paper is clear, the PEG algorithm presented is clever and novel, and the code release suggests that everything will be reproducible. The paper is already quite polished and does not require too much editing for a final copy, although some of the citations (e.g. top of page 2) seem misformatted. As mentioned previously, my only concern with the quality of the paper is with the rigour of the experimental results, which I think can be addressed through increased evaluations (smaller error bars) and a bit more analysis (particularly into 3-block stacking).

**Strength And Weaknesses:**

The paper's strengths include clear writing, interesting figures, and an effective new general purpose algorithm for goal-conditioned reinforcement learning. It effectively highlights differences between the new algorithm and baselines in (exploratory) goal setting approaches, with good explanations on how these differences ultimately contribute to better performance. The algorithm is well explained, and ablation tests validate the increased complexity of certain design choices. The description of future work and limitations of the current PEG algorithm (specifically regarding

In my opinion, the main weakness of the paper is the significance and standardization of the experimental results. The error bars on the results in figure 4 are borderline convincing that PEG is a significant step up from the baselines, particularly in the first 3 experiments. The differences highlighted in Figures 5 and 6 help to explain the marginally increased performance of PEG over baselines in those experiments, but there is an omission of good analysis into the deficiency of baseline policies relative to PEG in the 3-block stack experiment. Furthermore, the paper notes that evaluation goals were picked to "require extensive exploration" -- more work needs to be done to justify why this makes the test environments more representative of real-world problems, and why is does not simply bias the experimental results towards PEG (which is explicitly designed for this behaviour). I think that the "3-block stack" paragraph of subsection 4.3 could be improved through the use of figures and other analysis to demonstrate issues in the baselines -- I also believe that the use of the exclamation and rhetorical question in this paragraph ("PEG experiences... throughout training!" and "Did the... Explore-Phase?") are inappropriate and more passive language should be used in this section (particularly because the first statement is critical of prior works).

I would also like to see some discussion of value of information/exploration methods in the related works, since this approach of identifying states with "high-exploratory value" seems related to other works in the areas of quantifying the similarity of states and other analysis from general exploration in POMDPS.  And, I would like a bit more rigour in the implementation details (e.g., "optimization only takes a few seconds" on what kind of hardware? is this significantly faster/slower than X?).

**Summary Of The Paper:**

This paper presents the "Planning Exploratory Goals" algorithm, which identifies goal states that are likely to result in many novel observations when used to initialize an subsequent explore-phase upon reaching a termination condition en route to the aforementioned goal. The algorithm overcomes prior issues, such as working with poor quality starting policies and unreachable (exploratory) goals, by taking the agent's learned transition ("world") model into account. This enables it to effectively explore in many goal-conditioned reinforcement learning scenarios.

**Summary Of The Review:**

This is a strong paper that improves the state of the art in goal conditioned reinforcement learning in a wide range of problems. The algorithm is interesting and empirically validated, and I think it will inspire good follow-up work and discussion in the community.

---

> ### Author Response · Authors · 2022-11-12
> **Author Response Part 1/2 (11/11/22)**
>
> Thank you for these detailed comments!
> > …my only concern with the quality of the paper is with the rigour of the experimental results, which I think can be addressed through increased evaluations (smaller error bars) and a bit more analysis (particularly into 3-block stacking).
>
> > the paper notes that evaluation goals were picked to "require extensive exploration" -- more work needs to be done to justify why this makes the test environments more representative of real-world problems, and why is does not simply bias the experimental results towards PEG (which is explicitly designed for this behaviour).
>
> > The error bars on the results in figure 4 are borderline convincing that PEG is a significant step up from the baselines, particularly in the first 3 experiments.
>
> > The differences highlighted in Figures 5 and 6 help to explain the marginally increased performance of PEG over baselines in those experiments…
>
> Sophisticated exploration approaches are naturally most required on the hardest exploration tasks, which is where they can be most effectively evaluated. This affects our experimental choices, both in terms of the environments we use, and the specific goals within each environment that we evaluate on:
>
> - **Largest gains on block-stacking, smaller gains elsewhere:** We evaluated PEG in 4 environments (PointMaze, Walker, AntMaze, 3-Block Stack), presenting them in order of increasing difficulty. PEG’s performance margin over baselines increases with exploration difficulty, and is highest in 3-block stacking, our most difficult experiment - only PEG is able to reach the goal. Gains are expectedly smaller in other settings, but as the reviewer suggests, we will seek to increase confidence in those gains through retraining PEG and baselines with more random seeds. These are computationally intensive experiments, but we will report our progress before the end of the response period.
> - **Why only hard exploration goals?** Within each environment, we seek to evaluate PEG and baselines on goals that would have required good exploration to learn well, so that we can best differentiate between these exploration approaches. As such, these goals are selected to advantage good exploration approaches in general, since that is the problem we tackle, and they are not in any way tailored to PEG. However, we do agree that evaluating on more goals may be informative, as we do below.
> - **Why do baselines fare so poorly on block stacking?** On the block stacking environment, rather than only evaluating methods on the very hardest goal of 3-block stacking, we have now compared all methods on more tasks. We have defined three types of goals of increasing difficulty - picking up a single block, stacking two blocks, and then stacking three blocks. For each goal type, we set specific evaluation goals by varying the location of the pick / stack as well as the choice and ordering of the blocks. In this way, we create 3 distinct “picking” goals, 6 “2-block stacking” goals, and 6 “3-block stacking” goals (expanding from the lone 3-block stacking goal we had reported in the submission). We evaluate PEG and baselines (after 1M training steps) on these goals, and report the mean success rate and standard error over 5 seeds below:
> |  | Picking | 2-Stack | 3-Stack |
> |:---:|:---:|:---:|:---:|
> | PEG | 42.66% ± 10.97 | 17.66% ± 4.43 | 19.64% ± 2.75 |
> | MEGA | 50.65% ± 5.92 | 7.30% ± 4.75 |  5.97% ± 1.00 |
> | Skewfit | 58.65% ± 9.40 | 10.90 ± 4.40 | 6.96% ± 3.40 |
> | P2E |  43.96% ± 8.56 | 8.65% ± 2.64 | 7.06% ± 1.78 |
> | LEXA | 29.96% ± 2.10 | 2.73% ± 0.71 |  6.59% ± 1.82 |
>
> These results shed clearer light on our original block stacking results. Even though the baselines fail completely on the single 3-block stacking task reported in the submission, they do achieve non-trivial exploration behavior: this is clear from the fact that they can train policies for easier goals such as picking, and to some extent, 2-block stacking. Even on the expanded set of six 3-block stack goals, baseline performance remains poor (highest 7% compared to PEG 20%). PEG performs the best on both 2-stack and 3-stacks, showing that it explores these harder-to-reach stacking states more than the baselines.
>
> As a sidenote, we observe that the poorer exploration approaches fare marginally better on the easiest task, single-block picking. We believe that this is because for all methods, policies are trained on goal states sampled from the replay buffer (Step 7 of Algorithm 1). PEG explores more diverse states, thus the PEG goal-conditioned policy is required to train on many more diverse goals than MEGA or Skewfit, and policy optimization on this larger task space is more difficult. Instead, the MEGA or Skewfit policies can specialize very well to block-picking, ignoring the other harder tasks that are less represented in their replay buffers.

---

> > ### Author Response · Authors · 2022-11-12
> > **Author Response Part 2/2 (11/11/22)**
> >
> > Finally, we plan to extend this experiment further by ​​evaluating all methods on these goals during various stages of training (rather than only at the end), as well as evaluating all methods on non-hand-picked goals by uniformly sampling feasible goals from the goal space. See also our response to Reviewer rUvT, where we found PEG to perform well on uniformly sampled goals, while baselines fell short on hard-exploration goals.
> >
> > > I would also like to see some discussion of value of information/exploration methods in the related works, since this approach of identifying states with "high-exploratory value" seems related to other works in the areas of quantifying the similarity of states and other analysis from general exploration in POMDPS.
> >
> > Thank you for pointing us to this relevant line of work. We plan to include the following text into the manuscript, please let us know if we are missing any other key references, or if we are missing a deeper connection.
> >
> > “PEG reasons about the intrinsic motivation / exploration rewards to be gained by commanding a goal-conditioned task policy to various goals in an unsupervised RL setting. Related at a high level, a class of supervised exploration approaches reason about *task reward* gains during exploration through the notion of "value of information" (VoI) [1] . The VoI describes the improvement in supervised task rewards of the current optimal action policy operating under a specified limit on state information. An RL agent may modulate exploration-vs-exploitation in supervised RL settings, by tuning this information limit: a higher limit leads to lower exploration and greater exploitation [2, 3]. VoI exploration approaches operate in the supervised setting, and reason about task rewards when training a fixed task policy; instead, PEG operates in the unsupervised setting and reasons about intrinsic motivation rewards when training a goal-conditioned policy in the Go-Explore framework.”
> >
> > [1] R. L. Stratonovich and B. A. Grishanin, "Value of information when an estimated random variable is hidden", Izvestiya of USSR Academy of Sciences Technical Cybernetics, vol. 6, no. 1, pp. 3-15, 1966.
> >
> > [2] I. J. Sledge and J. C. Príncipe, "Balancing exploration and exploitation in reinforcement learning using a value of information criterion," 2017 IEEE International Conference on Acoustics, Speech and Signal Processing (ICASSP), 2017, pp. 2816-2820, doi: 10.1109/ICASSP.2017.7952670.
> >
> > [3] Cogliati Dezza, I., Yu, A.J., Cleeremans, A. et al. Learning the value of information and reward over time when solving exploration-exploitation problems. Sci Rep 7, 16919 (2017).
> >
> > > And, I would like a bit more rigour in the implementation details (e.g., "optimization only takes a few seconds" on what kind of hardware? is this significantly faster/slower than X?).
> >
> > We have updated the implementation details section and appendix with more information on runtime and compute resources, and summarize below for convenience. Note that we do not claim to be faster than baselines in walltime or resources, rather that any computational overheads introduced by our approach are minimal compared to the rest of the MBRL pipeline.
> >
> > Specifically, the runtime and  resource usage between methods did not differ significantly, as everything is implemented on top of the backbone LEXA model-based RL agent. Runtime is dominated by neural network updates of the policies and world model, not the goal selection routine defined by each method. Each seed was run on 1 GPU (Nvidia 2080ti or Nvidia 3090) and 4 CPUs, and required ~11GB of GPU memory.
> >
> > Total training time by experiment (roughly same for all methods)
> > |  | Total Runtime (Hours) | Episodes | Episode Length | Seconds per Episode |
> > |:---:|:---:|:---:|:---:|:---:|
> > | Point Maze | 16 | 12000  | 50 | 4.8 |
> > | Walker | 16 | 5000  | 150 | 11.52 |
> > | AntMaze | 48 | 2000  | 500 | 86.4 |
> > | 3-Block Stack | 48 | 6666 | 150 | 25.9 |
> >
> > We benchmarked the goal selection procedure for PEG and the other goal selection baselines in the block stacking environment and recorded the average wall clock time.
> >
> > |  | Seconds / Episode |
> > |:---:|:---:|
> > | PEG: | 0.51 |
> > | MEGA: | 0.48 |
> > | SkewFit: | 0.46 |
> >
> > We can see that there is little difference in speed between methods, and the overhead introduced by PEG goal selection is minimal to LEXA, and competitive with other goal selection baselines. Because LEXA does not select goals, it finishes up to an hour earlier than goal setting methods in our experiments. As a sidenote, the times reported above are amortized over episodes, since we compute a set of 50 goals every 50 episodes for all methods, rather than computing 1 goal per episode.
> >
> > Finally, thank you for catching the minor typos in citations; we have fixed them.

---

> > > ### Author Response · Authors · 2022-11-17
> > > **Reviewer 5sEN Followup (11/16/22)**
> > >
> > > Dear reviewer 5sEN, thank you again for your feedback. Per your comments (and those of reviewer rUvT) regarding more evaluations, we have now posted a global response with new results that point more clearly to PEG’s gains over the baselines.
> > >
> > > Separately, as the open discussion period draws to a close, we wanted to check back to see whether you have any remaining concerns. Your review pointed to two key concerns: the significance of PEG’s gains over baselines, and missing qualitative analysis of why baselines fared badly on block stacking. We believe that we have addressed both now, but would be grateful for any further feedback.

---

> > > > ### Comment · Reviewer_5sEN · 2022-11-22
> > > > **No further concerns**
> > > >
> > > > Thank you for your detailed response and follow-ups in the global responses. I'm quite happy with the latest version and the additions made to address the concerns raised by me and my fellow reviewers. I continue to feel very positive about this paper and have no further concerns.

---

### Author Response · Authors · 2022-11-12
**Global Response (11/11/22)**

We would like to thank all the reviewers for their comments and efforts towards evaluating our paper.  We have addressed all key reviewer concerns in responses to each reviewer, including through additional experiments where requested. In addition, we are running further experiments to back up those responses.

In this post, we would like to address the concern of Reviewers Wr3C and p4aC about PEG's relationship to prior work. We believe that the way we originally presented the work may have led to some lack of clarity on the precise connection of PEG with prior works, and what the novel component in PEG is. We clarify this below.

## Our contribution and related works
- First, Section 2: Problem Setup and Background in the paper describes all the key relevant work that we directly build on, particularly Go-Explore and LEXA (We include a summary below this list). Section 5, originally titled Related Work, would have more appropriately been titled “Other Related Work”, and indeed, we have renamed it as such now.
- Next, Section 3 describes the key new contributions of PEG, a model-based goal-setting algorithm for encouraging exploration in goal-conditioned RL. The objective described there, and its model-based implementation details is all novel. In the original submission, the algorithm block presented our PEG goal-setting within the context of a model-based goal-conditioned RL algorithm, LEXA. In the interest of clarity, we have now broken this down into separate algorithm blocks, first presenting LEXA in Section 2 Algorithm 1, LEXA’s random goal-setting-based exploration procedure in Section 2 Algorithm 2, and finally, our PEG goal-setting-based exploration procedure that replaces it in Section 3 Algorithm 3 (compare this directly to Algorithm 2).
- Finally, we have made note of additional relevant work pointed out by reviewers, and differentiate these from PEG in individual responses. We will incorporate them into the main text, or appendices as recommended by reviewers and as permitted by space limits.

**What does PEG add on top of LEXA / Go-Explore?**

**Go-Explore**: For convenience, we now summarize what PEG (Section 3) adds to Go-Explore and LEXA (Section 2). Go-Explore collects episodes by commanding a goal-conditioned policy to reach a goal, and then using the terminal state of that policy as the launching point for an exploration policy. Importantly, there is no prescribed general approach to select goals that lead to good exploration.

**Our key contributions**: Against this backdrop, PEG offers a general goal command selection objective for Go-Explore-style exploration in the unsupervised setting, directly maximizing an exploration value function. Next, we show how learned world models permit an effective implementation of PEG, by adapting sampling-based planning algorithms that are often used for low-level action sequence planning.


**Where LEXA comes in**: Since we propose to implement PEG through a learned world model, it is natural to consider model-based RL (MBRL) approaches that already train such a model. Thus we implement PEG on top of LEXA, a representative recent MBRL approach. To explore the environment, in each episode, LEXA either commands the agent to a randomly picked goal within its replay buffer, or runs a purely exploratory policy from the initial state. Thus, exploration in LEXA involves neither the unsupervised generalization of Go-Explore (launching specialized exploration policies from the terminal states of a goal-conditioned task policy), nor careful exploration-maximizing goal selection within this context. PEG implemented on LEXA adds both these features, yielding large gains on hard exploration tasks.

**Exploration Baselines In Our Experiments:** Algorithm blocks 1, 2, 3 in the revised paper further elucidate these differences. In summary, PEG is a general goal setting objective that yields goals for Go-Explore exploration. PEG can be used with any model-based GCRL agent to enhance exploration; in this work we use LEXA. Our baselines all build on top of LEXA, but implement different exploration approaches. While PEG directly optimizes the exploration value of Go-explore episodes, our MEGA and Skewfit baselines also implement Go-explore, but pick goals based on state density objectives. We also have 2 baselines that do not use Go-explore: Vanilla LEXA and P2E. LEXA sets random goals as explained above, and P2E altogether foregoes goal-setting as a mechanism for exploration. We further ablate the components of PEG in Figure 7, showcasing gains from each. We have now updated Appendix A.2 to include side-by-side pseudocode blocks for all baselines.

Finally, we believe that presenting the background works (Section 2) separately from other relevant works (Section 5) improves the readability of our main text, but we will include an additional “Extended Related Works” section in the appendix that presents a more unified picture of all this related work.

---

### Author Response · Authors · 2022-11-17
**Update (11/16/22): Block Stacking & PointMaze Results With More Random Seeds**

Reviewer 5sEN suggested running more random seeds in our main experiments to increase confidence in PEG’s gains over baselines. Similarly, Reviewer rUvT commented on the high variance of our block stacking results in particular.

The updated Figure 4 in the manuscript now shows mean and standard error of success rates over 10 seeds (up from 5) for Point Maze and 3-Block Stacking, confirming that PEG’s gains over baselines are well beyond the error margins in each case. We thank the reviewers for this experiment suggestion; it has indeed more clearly established our claims.

We are similarly running experiments with more random seeds for the other environments, Walker and AntMaze, but these may not complete within the rebuttal period. We have had to prioritize owing to the computational demands of these experiments: for 4 environments, 5 methods per environment, 10 random seeds per method, and 1-2 GPU-days per random seed, we would require 200-400 GPU days, which are unavailable to us currently. We have therefore prioritized 3-Block Stacking, our hardest exploration setting, and Point Maze, the fastest experiment.

As of today, barring further requests from reviewers, we will prioritize reporting at least preliminary results for one other experiment before the end of the response period, namely, evaluating PEG and baselines on more goals for 3-Block Stacking and AntMaze, not just at the *end* of training (as we have already reported in our initial response to Reviewers 5sEN and rUvT dated 11/11/22), but *during* training to more clearly track when different methods discover different skills.

---

### Author Response · Authors · 2022-11-18
**Update (11/18/22): More extensive evaluation in the block-stacking environment**

As our final experiment update during the response period, we evaluate PEG and baselines on more goals in the block stacking environment, not just at the end of training (as we have already reported in our initial response to Reviewers 5sEN and rUvT dated 11/11/22), but during training, to stay consistent with the original plots in the paper.

Specifically, in the block stacking environment, recapping our previous post, we have defined three types of goals of increasing difficulty - picking up a single block (“Easy”), stacking two blocks (“Medium”), and stacking three blocks (“Hard”). For each goal type, we set specific evaluation goals by varying the location of the pick / stack as well as the choice and ordering of the blocks. In this way, we create 3 distinct Easy goals , 6 Medium goals , and 6 Hard goals. This permits a more extensive evaluation than using one lone 3-block stacking (Hard) goal, as we had reported in the submission.

In Appendix A.7 of the revised submission, we report the mean success and standard error over 10 seeds for each goal type, tracked through various stages of training. We highlight the key take-aways here.

First, the trends are consistent with the original plots; PEG remains clearly the strongest at 3-block stacking (Hard).

Second, the difficulty-based task grouping permits more interesting analysis: PEG performs competitively with the best approaches on the Easy goals, and its gains over baselines are larger for Medium and Hard goals where exploration is most required. On these harder categories of tasks, PEG is both quickest to achieve non-trivial performance (indicating the onset of exploration) and also achieves the highest peak performance.

Third, an interesting side note here is that better exploration tends to create greater challenges for multi-task goal-conditioned policy function approximation with a fixed capacity policy network. Around the 0.5M steps mark, when PEG starts to achieve Hard goals, it grows marginally weaker on Easy and Medium goals: multi-task learning on more tasks is harder. This kind of “forgetting” [1,2] is well-studied in the continual learning literature; and there may be interesting strategies we could borrow, such as expanding the policy network capacity over time [3,4].

Summing up, we have now performed a more extensive analysis of the 3-block stacking setting, as Reviewer 5sEN requests, confirming that PEG gains were not caused by evaluating on goals that incidentally advantaged PEG (addressing Reviewer 5sEN and rUvT’s concern), and also providing deeper insights into the timeline of different types of skill acquisition by PEG and other exploration approaches. We will integrate these insights into the paper and appendices as space permits.


[1] French, Robert M. "Catastrophic forgetting in connectionist networks." Trends in cognitive sciences 3.4 (1999): 128-135.

[2] Kirkpatrick, James, et al. "Overcoming catastrophic forgetting in neural networks." Proceedings of the national academy of sciences 114.13 (2017): 3521-3526.

[3] Rusu, Andrei A., et al. "Progressive neural networks." arXiv preprint arXiv:1606.04671 (2016).

[4] Yoon, Jaehong, et al. “Lifelong Learning with Dynamically Expandable Networks.”, International Conference on Learning Representations, (2018).

---

### Decision · Program_Chairs · 2023-01-20

**Decision:**

Accept: notable-top-25%

**Justification For Why Not Higher Score:**

The work is a good one but not the most impressive in this area.

**Justification For Why Not Lower Score:**

The work is an important contribution to the exploration problem.

**Metareview: Summary, Strengths And Weaknesses:**


This paper addresses the exploration problem for complex goal-reaching tasks within the goal-conditioned reinforcement learning (RL) paradigm. The authors propose planning exploratory goals (PEG), which combines ideas from two existing approaches: LEXA and Go-Explore, and sets goals for each training episode to optimize an intrinsic exploration reward directly.

All reviewers are generally appreciative of the work. The work addresses an important problem, develops a novel idea, and shows significant empirical evidence supporting their method. Hence, it meets the bar of this conference.

One issue with model-based methods is the computational cost and resource usage. Hence, including some discussions on reducing the computational cost of these methods in general and specifically for the proposed method would greatly benefit the community.

Another recommendation is to include the pseudocode for the interaction loop that generates the learning curves. Algorithms 1 through 3 are appreciated, but they need to indicate where the actual interactions with the environment happen clearly. Including the pseudo-code of Go-Explore will help a lot with that. Otherwise, the work may remain inaccessible to those not working on model-based RL.


**Note From Pc:**

if the above contains the word "oral" or "spotlight" please see: "oral" presentation means -> notable-top-5% and "spotlight" means -> notable-top-25%. As stated in our emails, we are disassociating presentation type from AC recommendations